# NODE NUMBER AWARENESS REPRESENTATION FOR GRAPH SIMILARITY LEARNING

## ABSTRACT

This work aims to address two important issues in the graph similarity computation, the first one is the **Node Number Awareness Issue ($\mathbf{N^2AI}$)**, and the second one is how to accelerate the inference speed of graph similarity computation in downstream tasks. We found that existing Graph Neural Network based graph similarity models have a large error in predicting the similarity score of two graphs with similar number of nodes. Our analysis shows that this is because of the global pooling function in graph neural networks that maps graphs with similar number of nodes to similar embedding distributions, reducing the separability of their embeddings, which we refer to as the $\mathbf{N^2AI}$. Our motivation is to enhance the difference between the two embeddings to improve their separability, thus we leverage our proposed **Different Attention (DiffAtt)** to construct **Node Number Awareness Graph Similarity Model ($\mathbf{N^2AGim}$)**. In addition, we propose the **Graph Similarity Learning with Landmarks ($\mathbf{GSL^2}$)** to accelerate similarity computation. $GSL^2$ uses the trained $N^2AGim$ to generate the individual embedding for each graph without any additional learning, and this individual embedding can effectively help $GSL^2$ to improve its inference speed. Experiments demonstrate that our $N^2AGim$ outperforms the second best approach on Mean Square Error by 24.3%(1.170 vs 1.546), 43.1%(0.066 vs 0.116), and 44.3%(0.308 vs 0.553), for AIDS700nef, LINUX, and IMDBMulti datasets, respectively. Our $GSL^2$ is at most 47.7 and 1.36 times faster than $N^2AGim$ and the second faster model. Our code is publicly available on https://github.com/iclr231312/N2AGim.

## 1 INTRODUCTION

Graph similarity computation is a fundamental problem for graph-based applications, e.g., graph data mining, graph retrieval, and graph clustering (Kriege et al., 2020; Ok & Korea, 2020). **Graph Edit Distance (GED)**, which is defined as the least number of graph edit operators to transform graph $G_i$ to graph $G_j$, is one of the most popular graph similarity metrics (Gao et al., 2010; Neuhaus et al., 2006; Bougleux et al., 2015). The graph edit operators are insert or delete a node/edge, or relabel an edge. Unfortunately, the exact GED computation is NP-Hard in general (Zeng et al., 2009), which is too expensive to leverage in the downstream tasks.

Recently, many Graph Neural Networks (GNNs) based graph similarity computation algorithms have been proposed to compute the GED in a faster manner (Bai et al., 2019; 2020; Li et al., 2019; Ling et al., 2021; Bai & Zhao, 2021; Wang et al., 2021). The GNN-based algorithms transform the GED value to a similarity score and use an end-to-end framework to learn to map the given two graphs to their similarity score. As a general framework, the Siamese neural network can be used to aggregate information on each graph, while the feature fusion module can be used to capture the similarity between them, and the Multi-layer Perceptron (MLP) is then leveraged for the regression.

However, the existing popular graph similarity models become very inaccurate in predicting the similarity of two graphs with similar number of nodes, as shown in Fig 1. It is clear that the MSE of all four models becomes large as the difference in the number of nodes in the two graphs becomes smaller. In order to better understand this issue, we present in Section 3 a theoretical analysis of the most widely used modules in the graph similarity models from a statistical viewpoint. As shown in Fig 2(a)-(e), our conclusion is that all global pooling functions, also called graph readout functions, map graphs with similar number of nodes to similar embeddings, which reduces the separability

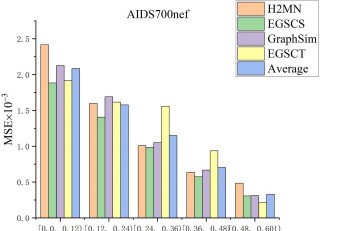 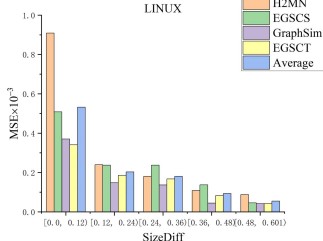 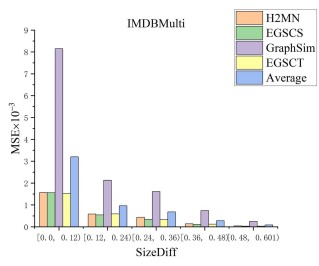

Figure 1: Histogram of the Mean Square Error (MSE) of the existing graph similarity models on three datasets at different level of SizeDiff. The SizeDiff represents the percentage difference in the number of nodes and is defined as $SizeDiff(G_1, G_2) = |N_1 - N_2|/max(N_1, N_2)$, where $N_i$ is the number of nodes in $G_i$. It is clear that all models have a larger MSE when the SizeDiff is smaller, i.e. when the number of nodes in the graph pair is similar.

between embeddings and leads to a large MSE for the models in predicting the similarity of two graphs with similar number of nodes. We refer to this issue of indistinguishable embeddings of graphs with similar number of nodes as the **Node Number Awareness Issue (N$^2$AI)**.

Our motivation to address the N$^2$AI is to focus more on the differences between two similar embeddings during the learning process, and we propose the **Different Attention (DiffAtt)** to construct our **Node Number Awareness Graph Simialrity Model (N$^2$AGim)**. DiffAtt is simple in architecture, and can be added as a plug-and-play module to any global pooling method. Our evaluations on three datasets (Section 5) demonstrate that the models with different pooling methods achieve a significant improvement after using DiffAtt. Moreover, our N$^2$AGim achieves state-of-the-art performance compared to the popular GNN-based graph similarity models, e.g., better about on average 33.3%(0.515 vs 0.772), 51.4%(0.515 vs 1.059) on Mean Square Error (MSE) than EGSCT (Qin et al., 2021) and GraphSim (Bai et al., 2020), respectively.

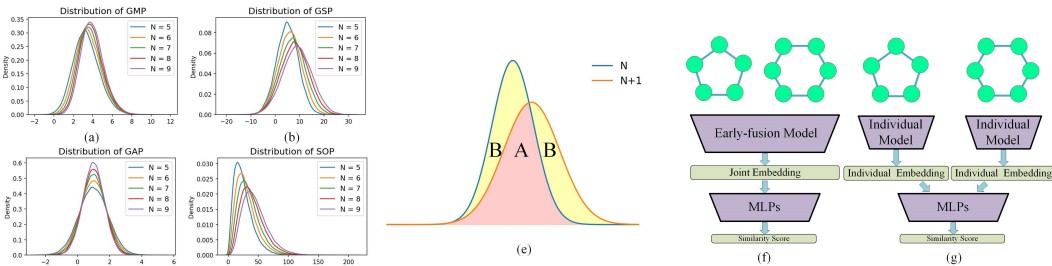

Figure 2: (a)-(d) Distributions of output from different global pooling functions with $N$ nodes, which show that all global pooling functions map graphs with similar number of nodes to similar distributions. See Section 3 for details. (e) Illustration of the N$^2$AI, i.e., the distribution of the embeddings of two graphs with similar number of nodes is indistinguishable. Region A represents where two distributions overlap, while B is the opposite. Our aim is to enhance the information in B to address the N$^2$AI. (f)-(g) Illustration of the Early Fusion Model (EFM) and Individual Embedding Model (IEM).

Another issue of interest in the field of graph similarity learning is to accelerate the inference speed of graph similarity models in downstream tasks. Qin et al. (2021) divided the graph similarity models into two categories, one is the Early Fusion Model (EFM), shown in Fig 2(f), which performs feature fusion at an early stage to achieve high accuracy but slow inference, and the other one is the Individual Embedding Model (IEM), shown in Fig 2(g), which generates an individual embedding for each graph and then performs fusion. This model is fast but achieves low accuracy. The existing solution (Qin et al., 2021) uses a special designed Knowledge Distillation (KD) paradigm to leverage an EFM teacher to improve the individual embeddings genergaed by the IEM student. However, motivated by Balcan et al. (2008), we propose a faster and more accurate IEM called **Graph Similarity Learning with Landmarks (GSL$^2$).** In GSL$^2$, a subset of graphs, called **landmarks** $\mathcal{S}$, are selected, and then each graph $G$ is represented as a vector $\boldsymbol{u}_G = [GED(G, \hat{G}_1), \cdots, GED(G, \hat{G}_m)]^T$, where $\hat{G} \in \mathcal{S}$. Finally, an MLP is learned to map the concatenation of the embeddings of the two graphs to their GED target. Instead of learning the embeddings on the graph data, our GSL$^2$ uses *an already trained graph similarity model* to directly generate an individual embedding for each graph, and this individual embedding can effectively improve the inference speed of GSL$^2$. To sum up, the contributions of this paper can be summarized as follows:

- We found that the existing graph similarity models have a relatively large error in predicting the actual similarity of two graphs with similar number of nodes, because the global pooling function maps graphs with similar number of nodes into two distributions that are similar, which we refer to as $\mathbf{N^2AI}$, thus reducing the performance of the graph similarity learning.
- A novel GNN-based graph similarity model, named $\mathbf{N^2AGim}$, is proposed. Our $N^2$AGim achieves excellent results in the graph similarity learning task by leverage the proposed **DiffAtt** to effectively address the $N^2$AI.
- In order to speed up the inference of graph similarity models, we propose the $\mathbf{GSL^2}$. The $GSL^2$ directly represents each graph as a vector, where each component is the GED value between the graph and a landmark. $GSL^2$ then learns the target GED values based on these graph representations.
- Experimental results show that our $N^2$AGim achieves the state-of-the-art performance, while our $GSL^2$ achieves a good accuracy and inference speed to efficiently handle downstream tasks.

## 2 PRELIMINARIES

### 2.1 GRAPH NEURAL NETWORKS

Graph data $\mathcal{G}$ can be viewed as a pair of adjacency matrix $A \in \{0,1\}^{N \times N}$ and a node feature matrix $X \in \mathbb{R}^{N \times C}$. $N$ is the number of nodes in the graph, and $C$ is the dimension of the initial node features. Node $i$ and node $j$ have an edge if and only if $A_{i,j} = 1$. Considering $X = [x_1, x_2, ..., x_N]^T$, a Message Passing Neural Network (MPNN) layer is defined as (Fey & Lenssen, 2019) : $x_i^{(k)} = \gamma^{(k)}(x_i^{(k-1)}, \square_{j \in \mathcal{N}(i)} \phi^{(k)}(x_i^{(k-1)}, x_j^{(k-1)}, e_{i,j}))$, where $x_i^{(k)} \in \mathbb{R}^{C_k}$ is an embedding of node $i$ at the $k^{th}$ layer, and $\phi$ performs a differentiable transform on each node or edge. $\square$ is an aggregate function to aggregate the transformed attributes of nodes and their neighbors. $\mathcal{N}(i)$ denotes the neighbors of node $i$ and $e_{i,j}$ is the edge feature from node $i$ to node $j$. $\gamma$ is a differentiable function to update the node embeddings. Following the idea of MPNN, several GNNs and their variants have been proposed to deal with graph mining tasks, e.g., Kipf & Welling (2016) and Velickovic et al. (2017). One of the vital works is the Graph Isomorphism Network (GIN) (Xu et al., 2018), which is at most as powerful as the Weisfeiler-Lehman (WL) graph isomorphism test (Leman & Weisfeiler, 1968), and it is defined as : $x_i' = h_\Theta \left( (1 + \epsilon) \cdot x_i + \sum_{j \in \mathcal{N}(i)} x_j \right)$, where $h_\Theta$ is an MLP. We believe that this representation ability is effective in addressing $N^2$AI. Therefore, we leverage the GIN layer as the backbone to construct our $N^2$AGim.

### 2.2 DEEP GRAPH SIMILARITY LEARNING

The graph similarity problem is defined as: given two graphs $G_i$ and $G_j$ with their similarity metric, the graph similarity models learn a function that maps the two graphs to their similarity metric. The Graph Matching Network (GMN) (Li et al., 2019) is the first deep graph similarity model, which computes the similarity between two given graphs by a cross-graph attention mechanism. Bai et al. (2019) turned the graph similarity task into a regression task. They not only proposed widely used graph similarity datasets, but also leveraged the GCN layers and self-attention-based fusion to design SimGNN. Further, in their later work (Bai et al., 2020), the proposed GraphSim directly learns the similarity based on the node-level interaction of the two given graphs. By leveraging a trained SimGNN to guide the search space of the A* algorithm, GENN-A* (Wang et al., 2021) achieves the best-in-class performance, *but needs too long inference time on the test data, i.e., 290.1 hours to solve the GED computation on AIDS700nef dataset (Qin et al., 2021). Considering that GENN-A* is too time-consuming in practice, we do not compare our proposed methods with it in our evaluations.* In order to achieve a faster speed, Qin et al. (2021) proposed a Knowledge Distillation (KD) paradigm to improve the individual graph embeddings generated by the student model.

However, we found that none of these existing graph similarity models are designed to address the $N^2$AI. In order to address the $N^2$AI, our $N^2$AGim leverages the GIN layers and the proposed DiffAtt to enhance the differences between the embeddings of two graphs and therefore achieves the state-of-the-art performance on benchmark datasets. Compared to EGSCS (Qin et al., 2021), our $GSL^2$ directly generates an individual graph representation for each graph, and the only learnable

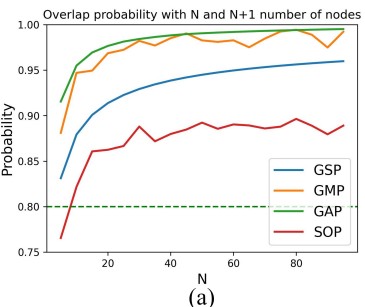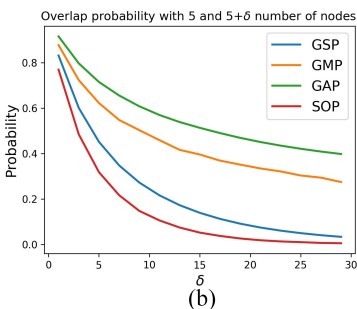

Figure 3: Overlapping probability of graph embedding with $N$ and $N + \delta$ number of nodes. (a) shows the overlap probabilities with different $N$ when the $\delta$ is 1; (b) shows the overlap probabilities for different $\delta$ when the $N$ is 5.

parameter of our GSL$^2$ is a simple MLP. Besides, the evaluation also shows that our GSL$^2$ has a higher accuracy and faster inference speed than EGSCS.

## 3 NODE NUMBER AWARENESS ISSUE (N$^2$AI) ANALYSIS

Here, we provide a formal theoretical analysis of N$^2$AI and reveal the reasons for its existence. GNNs usually generate the embeddings of a graph by multi-layer GNN aggregation layers and global pooling methods. The GIN is known for having at most as powerful as WL-Test, that is, to distinguish whether two graphs are isomorphic, which indicates that GIN is effective in distinguishing graphs with similar node numbers and address the N$^2$AI. Hence, we focus on the impact of widly used different global pooling methods on the N$^2$AI, including the one order statistical methods, i.e., Global Sum Pooling (GSP), Global Max Pooling (GMP), Global Average Pooling (GAP) and the second order statistical methods, i.e., Second Order Pooling (SOP) (Wang & Ji, 2020).

Let's assume that the node feature matrix output by the graph neural network layers is $X = [\boldsymbol{v}_1, \boldsymbol{v}_2, \cdots, \boldsymbol{v}_C]$, where $\boldsymbol{v}_i \in \mathbb{R}^N$ is the feature on the $i$th channel. We model all variables in $X$ as i.i.d random variables that follow a Gaussian distribution $\mathcal{N}(\mu, \sigma^2)$, where $\mu > 0$. The one order statistical methods $\mathcal{F}$ is used to convert $\boldsymbol{v}_i$ into a single value $g = \mathcal{F}(\boldsymbol{v}_i)$, which outputs a fix-sized vector, and the second order statistical methods convert $\boldsymbol{v}_i$ and $\boldsymbol{v}_j$ as a single value $g = \mathcal{F}(\boldsymbol{v}_i, \boldsymbol{v}_j)$, which outputs a fix-sized matrix. Here, we learn the N$^2$AI by studying whether a pooling method $\mathcal{F}_i$ can appropriately distinguish between $X$ with number of nodes $N$ and $N + \delta$, i.e., the differentiation between the two distributions $p(g|N, \mathcal{F}_i)$ and $p(g|N + \delta, \mathcal{F}_i)$, where $\delta$ denotes the difference between the number of nodes. We first assume that $X$ obeys $\mathcal{N}(1, 4)$ and show the output distribution of different pooling functions for different number of nodes in Fig 2(a)-(d). Intuitively, all four global pooling methods have a lot of overlap in terms of output distribution when the number of nodes is similar, and less overlap in terms of distribution when the number of nodes is very different.

We further define the probability of this overlap with the following equation:

$$O(\mathcal{F}_i, N, \delta) = \int \frac{min\{p(g|N, \mathcal{F}_i), p(g|N + \delta, \mathcal{F}_i)\}}{max\{p(g|N, \mathcal{F}_i), p(g|N + \delta, \mathcal{F}_i)\}} dg, \qquad (1)$$

where $O(\mathcal{F}_i, N, \delta)$ denotes the proportion of the overlapping area that occupies the total area of $\mathcal{F}_i$ with $N$ and $N + \delta$ nodes. Obviously, the outputs of the GSP and GAP obey the Gaussian distributions $\mathcal{N}(N\mu, N\sigma^2)$ and $\mathcal{N}(\mu, \frac{\sigma^2}{N})$, respectively, but it is difficult to obtain the distributions that the GMP and the SOP satisfy. Therefore, we perform a large number of randomized experiments and leverage the Kernel Density Estimation (KDE) to obtain an approximate distribution for the GMP and SOP. The overlapping probabilities of the four global pooling methods are shown in Fig 3.

From Fig 3(a), it is clear that most of the global pooling methods overlap more than 80% of the area of the distribution with $N$ and $N+1$ nodes, which means that existing graph similarity networks have difficulty distinguishing graphs with similar number of nodes in the output distribution, thus leading to the N$^2$AI. According to Fig 3(b), the probability of overlap between embedding distributions decreases as the difference in node counts increases. A way to address N$^2$AI is to make the graph similarity model focus on the differences between the two embeddings. Inspired by this, we propose the **DiffAtt** to enhance the difference between two embeddings generated by the above four global

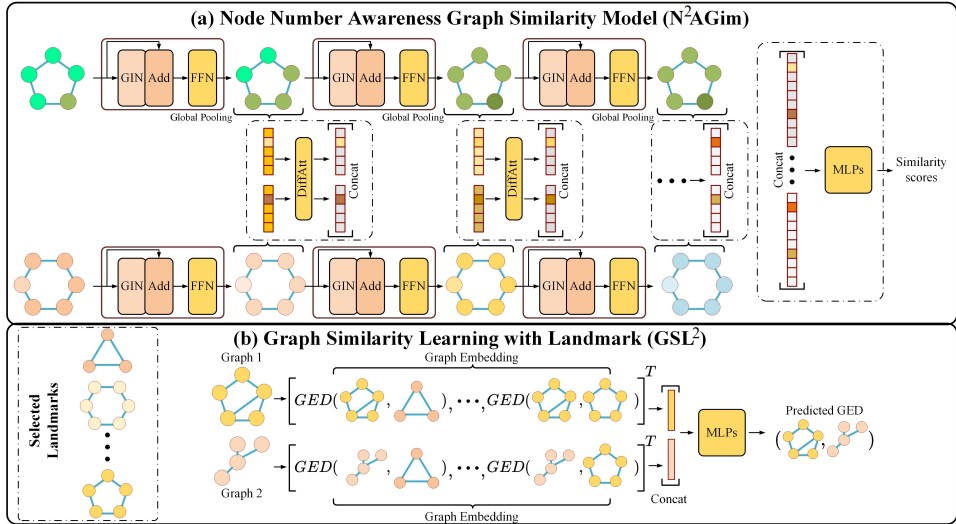

Figure 4: (a) $N^2$AGim first uses the multi-scale GIN layers to aggregate the information in the graph, then DiffAtt for feature fusion, and finally MLP to predict the similarity scores. (b) $GSL^2$ generates individual embeddings for each graph by calculating the GED values between them and landmarks, and then uses MLP to map the individual embeddings of the two graphs to their GED values.

pooling methods. The evaluation on benchmark datasets demonstrates that our DiffAtt brings a huge performance improvement to the graph similarity models.

## 4 PROPOSED METHODS

### 4.1 NODE NUMBER AWARENESS GRAPH SIMILARITY MODEL ($N^2$AGIM)

Our proposed $N^2$AGim consists of three stages: Multi-Scale GIN layers, Different Attention based feature fusion and the MLP regressor. Fig 4(a) shows a block diagram of our $N^2$AGim. On the following, we provide details of our proposed $N^2$AGim:

**Multi-Scale GIN layers.** Given a graph data $G = (A, X)$, where $A$ and $X$ are as defined in Section 2, the GIN layers, which can effectively address $N^2$AI because it is at most as powerful as WL-Test to distinguish whether two graphs are isomorphic or not, are leveraged as our backbone to update the node embeddings. All the MLPs in GIN have one linear layer with the Layer Normalization (Ba et al., 2016) and ReLU activation function. Besides, we apply the residual connections (He et al., 2016) and an additional FeedForward Neural Network (FFN) to enhance the node embeddings. We stack 3 GIN layers to aggregate multi-scale information of the node's neighbors. After each GIN layer, a one order statistical pooling method is applied to generate the graph embeddings.

**Different Attention based feature fusion.** We propose **Different Attention (DiffAtt)** to enhance the difference between the embeddings to address the $N^2$AI and obtain a joint embedding by fusing features of the two graph-level embeddings at each layer. Given the graph embeddings $\boldsymbol{h}_i^{(k)}$ and $\boldsymbol{h}_j^{(k)}$ at the $k$th layer, the DiffAtt is defined as :

$$Att^{(k)} = Softmax(MLPs^{(k)}(|\boldsymbol{h}_i^{(k)} - \boldsymbol{h}_j^{(k)}|))$$
$$u_{G_i}^{(k)} = flatten(Att^{(k)} \odot \boldsymbol{h}_i^{(k)}), u_{G_j}^{(k)} = flatten(Att^{(k)} \odot \boldsymbol{h}_j^{(k)}) \tag{2}$$

where $\boldsymbol{u}_{G_i}^{(k)} \in \mathbb{R}^C$ is the enhancement embeddings of $G_i$, the $flatten(\cdot)$ denotes the flatten operation, and $\odot$ denotes the Hadamard Product. It is evident that DiffAtt can give greater weight to large differences between the two embeddings and dynamically capture the differences that really matter with learnable parameters, which can effectively increase the separability of two graph embeddings with similar or even same number of nodes, thus effectively addressing $N^2$AI. Next, we concatenate the two enhancement embeddings as their joint embeddings as $\boldsymbol{u}_{G_i,G_j}^{(k)} = concat([\boldsymbol{u}_{G_i}^{(k)}, \boldsymbol{u}_{G_j}^{(k)}])$. Fi-

nally, we concatenate all the joint embeddings $\boldsymbol{u}_{G_i,G_j}^{(k)}$ at different layers to obtain a multi-scale joint embedding as $\boldsymbol{u}_{G_i,G_j} = concat([\boldsymbol{u}_{G_i,G_j}^{(0)}, \cdots, \boldsymbol{u}_{G_i,G_j}^{(3)}])$.

**MLP regressor.** A two-layer MLP is then applied to map $\boldsymbol{u}_{G_i,G_j}$ to the similarity scores. In the graph similarity task, the normalization GED is defined as $nGED(G_1, G_2) = \frac{GED(G_i,G_j)}{(|N_i|+|N_j|)/2}$, and the ground truth similarity score is defined as $exp(-nGED(G_i, G_j))$, which is in the range of (0,1]. We adopt the **Mean Square Error (MSE)** as the loss function to train N$^2$AGim.

## 4.2 GRAPH SIMILARITY LEARNING WITH LANDMARKS (GSL$^2$)

The graph similarity task inherently requires a deep fusion of the features of two graphs at the early stage and then learns from the joint embeddings to predict the similarity score, as shown in Fig 2(f). This makes it difficult to extract the individual embedding of each graph, which leads to higher computational costs in practice (Qin et al., 2021). Qin et al. (2021) used a KD-paradigm to improve the individual embeddings generated by the student IEM. In contrast, we provide a novel IEM framework, called **Graph Similarity Learning with Landmarks (GSL$^2$)**, which directly generates the individual embeddings of each graph without additional learning.

**Theorem 1** *Let landmarks $\mathcal{S}$ denote an infinite set containing every graph, and $\boldsymbol{u}_G$ is the embedding of graph $G$, defined as: $\boldsymbol{u}_G = [GED(G, \hat{G}_1), \cdots, GED(G, \hat{G}_{Infinity})]^T$, where $\hat{G}_i \in \mathcal{S}$. The GED value between any $G_1$ and $G_2$ satisfies:*

$$GED(G_1, G_2) = \min_i\{\boldsymbol{u}_{G_{1_i}} + \boldsymbol{u}_{G_{2_i}}\} = \min_i\{GED(G_1, \hat{G}_i) + GED(G_2, \hat{G}_i)\}, \quad (3)$$

The proof of Theorem 1 is provided in the Appendix A. Theorem 1 illustrates that the GED values between two graphs can be calculated by their GED values with landmarks. However, Theorem 1 needs to satisfy two requirements, the first one is to obtain an *infinite $\mathcal{S}$* and the second one is a large number of calculations of the *exact* GED values of the graphs and landmarks, both of which are impossible to satisfy in a practical scenario. The first requirement can be approximately solved by *randomly* selecting $M$ graphs from the training graph set to form $\mathcal{S}$. For the second requirement, the approximate GED values between graphs and landmarks can be computed quickly using a graph similarity model, e.g., SimGNN, GraphSim, or N$^2$AGim, which is equivalent to adding noise to the generated $\tilde{\boldsymbol{u}}_G$. However, in practice, we find that the direct use of $\min_i\{\tilde{\boldsymbol{u}}_{G_{1_i}} + \tilde{\boldsymbol{u}}_{G_{2_i}}\}$ to approximate the GED target $GED(G_1, G_2)$ has a relatively large error due to the limited number of landmarks and the noise in $\tilde{\boldsymbol{u}}_G$. Therefore, we propose to use MLP to learn to map the two generated embeddings to their GED target.

An illustration of our GSL$^2$ is shown in Fig 4(b), and the details are as follows: **First**, a subset of the graphs, $\mathcal{S} = \{\hat{G}_1, \cdots, \hat{G}_M\}$, named **landmarks**, are *randomly* selected from the the training graph set. **Second**, any graph similarity model can be leveraged to efficiently obtain the individual embeddings for each graph by computing their GEDs to the landmarks. However, from the above analysis, we can see that reducing the noise in $\tilde{\boldsymbol{u}}_G$ can improve the prediction accuracy. Therefore, we leverage our N$^2$AGim, which achieves the state-of-the-art performance, to calculate GED values for all graphs with landmarks. However, we found that directly converting exponential similarity values to GED values caused significant errors, so we used the ATS$^2$ similarity metric to retrain N$^2$AGim, see the Appendix B for details. **Third**, we concatenate two individual graph embeddings together in a joint embedding and learn an MLP to map the joint embedding to their GED target.

## 4.3 COMPARISON OF OUR N$^2$AGIM AND GSL$^2$

**Accuracy.** N$^2$AGim can effectively address N$^2$AI by fusing the features of two graphs by DiffAtt at multiple scales, and thus can achieve better performance. However, GSL$^2$ uses N$^2$AGim to quickly generate an individual embedding with noise for each graph and learn from the noisy embeddings, so the performance will be lower than that of N$^2$AGim.

**Inference speed.** Given $q$ query graphs, the aim is to compute the similarity between all query graphs and the $p$ graphs which already exist in the database. Assume the time to compute the similarity of a pair of graphs is $T_N$ for N$^2$AGim and the time to compute the similarity of a pair of embeddings is $T_{MLP}$ for the MLP in GSL$^2$. Since N$^2$AGim, as an EFM, requires fusion of graph pairs to obtain joint embeddings at each layer, it has a computational time of $p \times q \times T_N$.

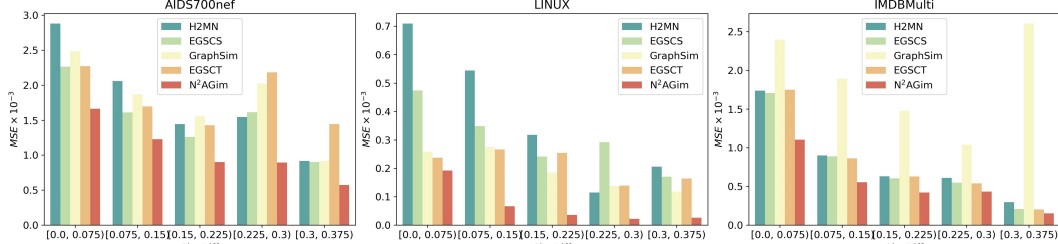

Figure 5: Visualisation of MSE for different models on test data with similar number of nodes. We split the test graph pairs with no more than 37.5% difference in the number of nodes into 5 bins for validation. Note that on graphs with a small number of nodes like AIDS700nef and LINUX, a difference of 7.5% SizeDiff represents a difference of approximately one node.

However, $GSL^2$, as a IEM, first generates the individual embedding of each graph using $N^2AGim$ and then predicts the similarity between the two embeddings using MLP, requiring a computation time of $(p + q) \times M \times T_N + p \times q \times T_{MLP}$, where $M$ is the number of landmarks. Since $GSL^2$ reduces the time complexity of computing joint embeddings and the inference speed of MLPs is generally lower than that of $N^2AGim$, i.e., $T_{MLP} \ll T_N$, the query tasks can be addressed more efficiently by $GSL^2$. Especially in industrial scenarios, graph data is usually preprocessed offline as the embedding. If all the graph embeddings are stored offline, the inference time of $GSL^2$ is just $p \times q \times T_{MLP}$. In summary, as an IEM, $GSL^2$ can be up to $T_N/T_{MLP}$ times faster than $N^2AGim$. The experimental results in Section 5.3 also demonstrate that $GSL^2$ can be up to 47.7 times faster than $N^2AGim$, which shows that $GSL^2$ can effectively address the similarity computation tasks.

## 5 EXPERIMENTS

In this section, we evaluate our proposals using the AIDS700nef, LINUX and IMDBMulti datasets provided by Bai et al. (2019) for the graph similarity learning and compare our method with other state-of-the-art methods. The statistics and details of these datasets and data processing are provided in Appendix B. Note that all the experiments are performed on a Linux server with Intel(R) Xeon(R) Gold 6226R CPU @ 2.90GHz and 8 NVIDIA GeForce RTX 2080Ti. The evaluation metrics we adopted are Mean Square Error (MSE) (in the format of $10^{-3}$), Spearman's Rank Correlation Coefficient ($\rho$) and Precision at 10 (p@10). All metrics with their meanings are listed in the Appendix. The $N^2AGim$ is evaluated using the PyTorch Geometric (Fey & Lenssen, 2019). We use Adam optimizer, a learning rate of 0.001, the batch size is set to 2000, and the hidden channel is set to 64. We run 200 epochs on the three datasets, and after running 150 epochs, we perform validation at the end of every epoch. Ultimately, the parameter that results in the least validation loss is chosen to perform the evaluations on the test data. We implement the $GSL^2$ using the PyTorch (Paszke et al., 2019), and the details can be found in our source code.

### 5.1 ABLATION STUDY

We perform ablation studies to show the influence of the DiffAtt in $N^2AGim$ and $GSL^2$ with different GED computation algorithms. For $N^2AGim$, we compare the performance of using and without using DiffAtt across the four global pooling functions mentioned. Besides, we compare our DiffAtt with the other popular attention based global pooling methods, i.e., Neural Tensor Network (NTN) (Bai et al., 2019), Embedding Fusion Network (EFN) (Qin et al., 2021), Global Soft Attention (GSA) (Li et al., 2015), Set2Set (Vinyals et al., 2015), Context Based Attention (CBA) (Bai et al., 2019), and Cross Context Based Attention (C2BA), under the same architecture. It is worth noting that most of the existing graph similarity models leverage the CBA to generate graph embeddings, e.g., Li et al. (2019); Bai et al. (2019); Qin et al. (2021); Zhang et al. (2021), which is defined as $\mathcal{F}_{CBA}(X) = \sum_{n=1}^{N} sigmoid(\boldsymbol{x}_n^T \boldsymbol{c})\boldsymbol{x}_n$, where $\boldsymbol{c}$ denotes the context information of the graphs. *C2BA is different from CBA only in that global context information $\boldsymbol{c}$ is from another graph in the graph pair.* For our $GSL^2$, we experiment with different graph similarity models. We also provide additional ablation experiments on the hyperparameter selection of our methods, including selecting different numbers of landmarks in $GSL^2$, etc. in Appendix G.

Table 1 demonstrates that our DiffAtt effectively improves 44 metrics out of 48 metrics of the four global pooling methods, especially giving a huge boost to 12 metrics on the IMDBMulti dataset,

Table 1: Results of the ablation study on using or without using our DiffAtt. Bold means the best. The Average denotes the average value over the three datasets. The ↑ denotes that the larger this indicator is, the better the performance, while the ↓ indicates the opposite.

| $\mathcal{F}_i$ | DiffAtt | AIDS700nef | | | LINUX | | | IMDBMulti | | | Average | | |
|---|---|---|---|---|---|---|---|---|---|---|---|---|---|
| | | MSE↓ | $\rho$↑ | P@10↑ | MSE↓ | $\rho$↑ | P@10↑ | MSE↓ | $\rho$↑ | P@10↑ | MSE↓ | $\rho$↑ | P@10↑ |
| GAP | ✗ | **3.004** | **0.819** | 0.486 | 0.457 | 0.987 | 0.970 | 0.753 | 0.866 | 0.878 | 1.405 | 0.891 | 0.778 |
| GAP | ✔ | 3.042 | 0.810 | **0.511** | **0.256** | **0.990** | **0.979** | **0.404** | **0.907** | **0.890** | **1.234** | **0.902** | **0.793** |
| GMP | ✗ | 3.051 | 0.817 | 0.501 | **0.396** | **0.988** | 0.970 | 0.386 | 0.871 | 0.875 | 1.278 | 0.892 | 0.782 |
| GMP | ✔ | **2.839** | **0.827** | **0.538** | 0.440 | 0.987 | **0.985** | **0.305** | **0.920** | **0.896** | **1.195** | **0.911** | **0.806** |
| SOP | ✗ | 3.039 | 0.811 | 0.496 | 0.261 | 0.991 | 0.973 | 0.776 | 0.892 | 0.863 | 1.359 | 0.898 | 0.777 |
| SOP | ✔ | **1.144** | **0.918** | **0.663** | **0.066** | **0.993** | **0.998** | **0.309** | **0.915** | **0.903** | **0.506** | **0.942** | **0.855** |
| GSP | ✗ | 1.396 | 0.903 | 0.624 | 0.116 | 0.992 | 0.983 | 0.392 | 0.859 | 0.884 | 0.635 | 0.918 | 0.830 |
| GSP | ✔ | **1.170** | **0.916** | **0.672** | **0.066** | **0.994** | **0.995** | **0.308** | **0.918** | **0.893** | **0.515** | **0.943** | **0.853** |

Table 2: Results of the ablation study of comparing our DiffAtt with other attention methods. Bold means the best, and † means the next best.

| | AIDS700nef | | | LINUX | | | IMDBMulti | | | Average | | |
|---|---|---|---|---|---|---|---|---|---|---|---|---|
| | MSE↓ | $\rho$↑ | P@10↑ | MSE↓ | $\rho$↑ | P@10↑ | MSE↓ | $\rho$↑ | P@10↑ | MSE↓ | $\rho$↑ | P@10↑ |
| NTN | 2.456 | 0.845 | 0.554 | 0.139 | 0.993† | 0.987 | 0.463 | 0.891 | 0.874 | 1.019 | 0.910 | 0.805 |
| GBA | 3.224 | 0.814 | 0.499 | 0.754 | 0.982 | 0.961 | 0.368 | 0.869 | 0.881 | 1.449 | 0.888 | 0.780 |
| Set2Set | 3.308 | 0.815 | 0.511 | 0.543 | 0.985 | 0.974 | 0.355 | 0.876 | 0.878 | 1.402 | 0.892 | 0.788 |
| CBA | 1.487 | 0.902 | 0.611 | 0.165 | 0.990 | 0.987 | 0.406 | 0.855 | 0.884 | 0.686 | 0.916 | 0.827 |
| C2BA | 1.269 | 0.911 | 0.662 | 0.094 | 0.993 | **0.995** | 0.439 | 0.887 | 0.875 | 0.601 | 0.930 | 0.844† |
| EFN | 1.249† | 0.912† | 0.644† | 0.078† | 0.993† | 0.990† | 0.315† | 0.902† | 0.891† | 0.547† | 0.936† | 0.842 |
| DiffAtt | **1.170** | **0.916** | **0.672** | **0.066** | **0.994** | **0.995** | **0.308** | **0.918** | **0.893** | **0.515** | **0.943** | **0.853** |

such as 46.3%(0.404 vs 0.753), 21.0%(0.305 vs 0.386), 60.2%(0.309 vs 0.776) and 21.4%(0.308 vs 0.392) on the four MSE metrics. This shows that DiffAtt has powerful generalization to effectively solve the N²AI of all pooling methods. In terms of the average results, SOP and GSP showed better results than GMP and GAP after the use of DiffAtt, which is the result of the smaller percentage of overlap area and greater distribution differences. Because of the higher performance and less computational cost of GSP, we finally chose it as the global pooling function in N²AGim.

Compared to other attention mechanisms, in Table 2, DiffAtt achieves the best results on all metrics under the same experimental setup and architecture, especially better than the EFN on average on three metrics 5.8%(0.515 vs 0.547), 0.7%(0.943 vs 0.936) and 1.3%(0.853 vs 0.842), respectively. As can be seen from the Table 3, the accuracy of GSL² increases as the generated GEDs are closen to the true GED values, which validates our analysis in Section 4.2 and shows that the performance of GSL² can be improved by using our N²AGim.

## 5.2 GRAPH SIMILARITY LEARNING

We compare our N²AGim and GSL² with a number of state-of-the-art methods for graph similarity learning tasks: GMN (Li et al., 2019), SimGNN (Bai et al., 2019), H2MN (Zhang et al., 2021), GraphSim (Bai et al., 2020), and EGSC (Qin et al., 2021). We note discrepancies in the MSE reported by various papers. For examples, the MSE reported in Bai et al. (2019), Bai et al. (2020) and Zhang et al. (2021) is $\frac{1}{2|D|}\sum_{i=1}^{D}(s-\hat{s})^2$, but Qin et al. (2021) reported MSE as $\frac{1}{|D|}\sum_{i=1}^{D}(s-\hat{s})^2$. To provide a consistent comparison, we use the MSE metric of the latter formula, and the results are shown in Table 4. Our N²AGim achieves the best performance in most of the cases. On AIDS700nef, the performance is improved by about 24.3%(1.170 vs 1.546 on MSE), 2.0%(0.916 vs 0.898 on $\rho$) and 3.5%(0.672 vs 0.649 on p@10) compared to EGSCS. On LINUX, our N²AGim achieves the

Table 3: Results of the ablation study of comparing different graph similarity models used in GSL². All graph similarity models were trained using ATS² and we transform the results to the exponential similarity scores and report it. The brackets represent the GED algorithm used in GSL² and the GT represents the ground truth GED values. $M$ is set to 60, 30 and 70 on three datasets, respectively.

| | AIDS700nef | | | LINUX | | | IMDBMulti | | | Average | | |
|---|---|---|---|---|---|---|---|---|---|---|---|---|
| | MSE↓ | $\rho$↑ | P@10↑ | MSE↓ | $\rho$↑ | P@10↑ | MSE↓ | $\rho$↑ | P@10↑ | MSE↓ | $\rho$↑ | P@10↑ |
| SimGNN | 3.151 | 0.827 | 0.397 | 0.752 | 0.983 | 0.921 | 3.722 | 0.934 | 0.826 | 2.542 | 0.915 | 0.715 |
| GraphSim | 1.824 | 0.889 | 0.562 | 0.283 | 0.991 | 0.979 | - | - | - | 1.053 | 0.94 | 0.771 |
| N²AGim | 1.184 | 0.917 | 0.675 | 0.071 | 0.994 | 0.989 | 0.341 | 0.973 | 0.9 | 0.532 | 0.961 | 0.855 |
| GSL²(SimGNN) | 2.187 | 0.873 | 0.518 | 0.402 | 0.987 | 0.975† | 0.668 | 0.949 | 0.852† | 1.086 | 0.936 | 0.782 |
| GSL²(GraphSim) | 1.824 | 0.883 | 0.505 | 0.210 | 0.990 | 0.971 | - | - | - | 1.017 | 0.936 | 0.738 |
| GSL²(N²AGim) | 1.470† | 0.905† | 0.604† | 0.074† | 0.994† | 0.995 | **0.510** | 0.971† | **0.869** | 0.685† | 0.957† | 0.822† |
| GSL²(GT) | **1.258** | **0.915** | **0.633** | **0.068** | **0.995** | **0.995** | 0.512† | **0.985** | 0.850 | **0.613** | **0.965** | **0.826** |

Table 4: Results of the graph similarity learning task. Bold means the best.

| | AIDS700nef | | | LINUX | | | IMDBMulti | | | Average | | |
|---|---|---|---|---|---|---|---|---|---|---|---|---|
| | MSE $\downarrow$ | $\rho\uparrow$ | P@10$\uparrow$ | MSE$\downarrow$ | $\rho\uparrow$ | P@10$\uparrow$ | MSE$\downarrow$ | $\rho\uparrow$ | P@10$\uparrow$ | MSE$\downarrow$ | $\rho\uparrow$ | P@10$\uparrow$ |
| GMN | 3.772 | 0.751 | 0.401 | 2.054 | 0.933 | 0.833 | 8.844 | 0.725 | 0.604 | 4.890 | 0.803 | 0.613 |
| SimGNN | 2.378 | 0.843 | 0.421 | 3.018 | 0.939 | 0.942 | 2.528 | 0.878 | 0.759 | 2.641 | 0.887 | 0.707 |
| GraphSim | 1.574 | 0.874 | 0.534 | 0.116 | 0.981 | 0.992 | 1.486 | 0.926 | 0.828 | 1.059 | 0.927 | 0.785 |
| H2MN | 1.826 | 0.881 | 0.521 | 0.210 | 0.990 | 0.975 | 1.178 | 0.913 | 0.889 | 1.071 | 0.928 | 0.795 |
| EGSCT | 1.601 | 0.901 | 0.658 | 0.163 | 0.988 | 0.994 | 0.553 | 0.938 | 0.872 | 0.772 | 0.942 | 0.841 |
| EGSCS | 1.546 | 0.898 | 0.649 | 0.293 | 0.984 | 0.978 | 0.581 | 0.935 | 0.857 | 0.807 | 0.939 | 0.828 |
| $N^2$AGim | **1.170** | **0.916** | **0.672** | **0.066** | **0.994** | **0.995** | **0.308** | 0.918 | **0.893** | **0.515** | 0.943 | **0.853** |
| GSL$^2$ | 1.470 | 0.905 | 0.604 | 0.074 | **0.994** | **0.995** | 0.510 | **0.971** | 0.869 | 0.685 | **0.957** | 0.822 |

Table 5: Results of inference time for each model. The suffix of '-R' means that the input is the raw query graph, while the suffix of '-F' means that the embeddings of the query graph are stored offline. All times reported below are in seconds.

| | SimGNN | $N^2$AGim | GraphSim | EGSCS-R | EGSCS-F | GSL$^2$-R | GSL$^2$-F |
|---|---|---|---|---|---|---|---|
| AIDS700nef | 5.106 | 9.245 | 4.383 | 4.383 | 0.975 | 3.874 | **0.718** |
| LINUX | 8.582 | 13.163 | 9.120 | 9.120 | 1.414 | 5.007 | **1.159** |
| IMDBMulti | 122.939 | 87.032 | 114.676 | 87.032 | 2.256 | 15.792 | **1.824** |

best performance in all three metrics, especially in MSE which is 43.1%(0.066 vs 0.116) better than the second best model, GraphSim. On IMDBMulti, $N^2$AGim achieves the best MSE and p@10 performance, but close not perform as well on $\rho$ (0.918 vs 0.938) than EGSCT. Although GSL$^2$ does not learn embeddings directly in the graph data, it achieves the state-of-the-art performance on three of the nine metrics on three datasets. Compared to the EGSCS, our GSL$^2$ achieved better performance in eight of the nine metrics on three datasets, demonstrating the powerful expressive ability of the generated embeddings in GSL$^2$. Compared to GSL$^2$, which learns on embeddings with noise, $N^2$AGim achieves a better performance, especially better by about 20.4%(1.170 vs 1.470), 1.2%(0.916 vs 0.905) and 11.3%(0.672 vs 0.604) on AIDS700nef. In addition, we visualised the MSE in test data with similar number of nodes for different models in Fig 5. *Compared to the other models, $N^2$AGim shows a significant improvement on graph pairs with similar number of nodes, demonstrating its effectiveness in addressing $N^2$AI.*

## 5.3 INFERENCE TIME

In this section, we provide a comparison of inference times for GSL$^2$ and the rest of the graph similarity models on test data. Our evaluation reflects real-world graph queries: we treat the training graph set as the graphs that already exist in the database and can be preprocessed, and the test graph set as the query graph. We calculate the similarity of a query graph to all graphs in the database at once to obtain the total query time, and all times are averaged over five tests. The results are shown in Table 5. By obtaining individual embeddings of the graphs offline, GSL$^2$-F comes out to be 12.9, 11.3 and 47.7 times faster than $N^2$AGim on the three datasets, respectively. Compared to EGSCS-F, GSL$^2$-F is 1.36, 1.22 and 1.24 times faster, respectively. This shows the potential of GSL$^2$ to efficiently compute graph similarity in realistic scenarios.

## 6 CONCLUSION

This paper addresses two issues in graph similarity tasks, one is the **$N^2$AI** and the other is the issue of improving the speed of graph similarity model inference for downstream tasks. By analysing the performance of popular graph similarity models, we show that graph similarity models have difficulty distinguishing the embeddings of two graphs with similar number of nodes, because the global pooling function maps graphs with similar number of nodes to similar embedding distributions, reducing the separability between embeddings. Therefore, **DiffAtt** is proposed to enhance the difference between two similar embeddings, thus the proposed **$N^2$AGim** achieves the state-of-the-art performance. To speed up the graph similarity computation, the **GSL$^2$** is proposed. Instead of learning embeddings in graph data, GSL$^2$ generates individual embeddings directly by a trained graph similarity model. Our analysis and experiments both demonstrate that such individual embeddings have a powerful expressive ability and can efficiently handle downstream tasks.

## 7 ETHICS STATEMENT

This work proposes two methods to address the real-time graph similarity tasks. Our proposed methods have a great potential for practical graph-based applications due to their high precision and high speed. Our methods also be applied to address any similarity problem between the graph data, e.g., the binary function similarity problem, which can be helpful for the software copyright issue. Therefore, we believe that our methods do not have any negative impact of the society but make positive impact of the society.

## 8 REPRODUCIBILITY STATEMENT

Our code is publicly available on https://github.com/iclr231312/N2AGim. We provide trained models and test code in our anonymous repository to help researchers quickly reproduce test results. Besides, we provide the source code for the training, including the hyperparameter settings and the fixed random seed we use to ensure our work is reproducible. Please get more details from our repository.

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

## A    PROOF OF THEOREM 1.

Here, we provide a proof of Theorem 1. The GED satisfies the triangle inequality, i.e., for any $G_1$ ,$G_2$ and $G_3$, there must exist:

$$GED(G_1, G_2) \leq GED(G_1, G_3) + GED(G_2, G_3). \tag{4}$$

In particular, when $G_3$ is isomorphic to a graph on the least-cost edit path between $G_1$ and $G_2$, there must exist:

$$GED(G_1, G_2) = GED(G_1, G_3) + GED(G_2, G_3). \tag{5}$$

Thus, assume there exists an infinite set $S$ containing every graph, any graph can be encoded as an infinitely long vector as:

$$\boldsymbol{u}_G = [GED(G, \hat{G}_1), GED(G, \hat{G}_2), \cdots, GED(G, \hat{G}_{Infinity})]^T, \tag{6}$$

where $\hat{G}_i \in \mathcal{S}$, so that for any two graphs $G_1$ and $G_2$, it exists:

$$GED(G_1, G_2) = \min_i(\boldsymbol{u}_{G_{1i}} + \boldsymbol{u}_{G_{2i}}). \tag{7}$$

## B    DATASETS AND PRE-PROCESSING

We perform an evaluation of our methods on AIDS700nef, LINUX and IMDBMulti datasets provided by Bai et al. (2019). Following is a brief overview of benchmark datasets:

1. **AIDS700nef** dataset contains 700 graphs from AIDS dataset which represent antivirus screen chemical compound, and all of them have 10 or less than 10 nodes.

2. **LINUX** dataset contains 1000 graphs selected from Wang et al. (2012), which represent Program Dependence Graph (PDG) generated by Linux kernel.

3. **IMDBMulti** dataset(Yanardag & Vishwanathan, 2015) contains ego-networks of actors/actresses, where nodes represent an actor/actress and edges indicate that these two actors/actress participated in the same movie.

For AIDS700nef and LINUX datasets, Bai et al. (2019) compute the GED of every graph pair using an algorithm named A*, and for IMDB datasets, the minimum of GED computed by three algorithms: Beam (Neuhaus et al., 2006), Hungarian (Riesen & Bunke, 2009) and VJ (Fankhauser et al., 2011), is considered as the ground truth. In order to enhance the node features of the graph, we concatenate the one-hot encoding of the node degree into its features on these three datasets. Note that, the GED metric is first normalized as $nGED = \frac{GED_{G_i, G_j}}{0.5 \cdot (|G_i| + |G_j|)}$, where $|G_i|$ represents the number of nodes in $G_i$. and then adopted a function $\lambda(x) = e^{-x}$ to transform to range (0,1]. We randomly split datasets into $60\%, 20\%, 20\%$ as training graph set $Tr$, validation graph set $V$, and testing graph set $Te$, respectively. We take the Cartesian product of $Tr$ labeled with their similarity scores as the the training set. The validation set (testing set) is defined as the Cartesian product of $Tr$ and $V$ ( $Te$ ) labeled with the ground truth. The training set is defined as $\{(G_i, G_j, s_{G_i, G_j}) | G_i \in Tr, G_j \in Tr\}$, where $s_{G_i, G_j}$ denotes the similarity score of $G_i$ and $G_j$, and the validation dataset and the testing dataset is $\{(G_i, G_j, s_{G_i, G_j}) | G_i \in Tr, G_j \in V\}$, $\{(G_i, G_j, s_{G_i, G_j}) | G_i \in Tr, G_j \in Te\}$, respectively.

However, we found in GSL$^2$ that directly converting the exponential similarity scores predicted by the graph similarity models to GED values can cause significant errors. Therefore, we used a new similarity score to train the graph similarity models, called **Adaptive Transform Similarity Scores (ATS$^2$)**, and transform the results back to exponential similarity score for comparison with other models at test time. The ATS$^2$ is defined as:

$$ATS^2(G_1, G_2) = 1 - \lg(nGED(G_1, G_2) + 1) / \lg(\max_{i,j}\{nGED(G_i, G_j)\} + 1). \tag{8}$$

Table 6: Statistics of all the datasets used in our experiments.

| Datasets | Graphs | Avg nodes | Avg edges | Pairs of testing graphs | Node attr |
|----------|--------|-----------|-----------|-------------------------|-----------|
| AIDS700nef | 700 | 8.9 | 17.6 | 58,800 | ✔ |
| LINUX | 1000 | 7.58 | 13.87 | 120,000 | ✗ |
| IMDBMulti | 1500 | 13 | 65.94 | 270,000 | ✗ |

## C  EVALUATION METRICS

The evaluation metrics that we adopted are the Mean Square Error (MSE), the Spearman's Rank Correlation Coefficient ($\rho$) (Spearman, 1961), and the Precision at 10 (P@10) (Bai et al., 2019). Moreover, we provide the results of the $\tau$ (Kendall, 1938) and P@20 metrics in the experiments in Appendix. The MSE metric can accurately calculate the distance between the predition results from the model and the ground truth, and $\rho$ and $\tau$ evaluate the matching between the global ranking result of the prediction results and the ground truth, while P@k is the intersection of the top $k$ results of the prediction and the ground truth.

## D  N²AI

Here, we provide a more detailed description of the N²AI in the popular graph similarity models. We grouped the testing set by the number of nodes in the graph and counted the MSEs. We normalized the MSEs with Min-Max Normalization, and visualized the results in Fig 6. It is clearly that the large MSE are all concentrated in locations with similar number of nodes, which reflects the prevalence of N²AI.

## E  N²AGIM WITH SECOND ORDER POOLING

Given the feature map $X^k$ at the $k$th layer, Second Order Pooling (SOP) is defined as:

$$H^{(k)} = \mathcal{F}(X^{(k)}) = (X^{(k)})^T \cdot (X^{(k)}) \tag{9}$$

where $H^{(k)} \in \mathbb{R}^{C \times C}$ is a fixed size matrix. For the SOP, we define the **DiffAtt** as :

$$
\begin{aligned}
Diff^{(k)} &= |H_i^{(k)} - H_j^{(k)}| \\
Att^{(k)} &= Softmax2D(Diff^{(k)} \otimes \Theta^{(\mathbf{k})} + bias^{(k)}) \\
\boldsymbol{U}_{G_i,G_j}^{(k)} &= Att^{(k)} \odot (H_i^{(k)} H_j^{(k)^T})
\end{aligned}
\tag{10}
$$

Where $\boldsymbol{U}_{G_i,G_j}^{(k)} \in \mathbb{R}^{C \times C}$ is the joint embeddings, and $\Theta^{(k)} \in \mathbb{R}^{C \times C}$ and $bias^{(k)} \in \mathbb{R}^{C \times C}$ are two learnable parameter. If the $\boldsymbol{U}_{G_i,G_j}^{(k)}$ is directly flattened into a vector for regression, it will result in a larger amount of parameters for the model. Therefore, motivated by the Bai et al. (2020), we apply four convolution layers with residual connections to learn the $\boldsymbol{U}_{G_i,G_j}^{(k)}$, and flatten the feature map to a vector $\boldsymbol{u}_{G_i,G_j}^{(k)}$ to reduce the parameters of the model. Finally, we concatenate all the joint embeddings $\boldsymbol{u}_{G_i,G_j}^{(k)}$ at different layers to obtain a multi-scale joint embedding as $\boldsymbol{u}_{G_i,G_j} = concat([\boldsymbol{u}_{G_i,G_j}^{(0)}, \cdots, \boldsymbol{u}_{G_i,G_j}^{(3)}])$.

## F  JOINT EMBEDDING VISUALISATION WITH DIFFATT AND WITHOUT DIFFATT

We also visualised the joint embeddings generated with and without DiffAtt with each of the three datasets using T-SNE, as shown in Fig 7. It is clearly that the joint embeddings generated with DiffAtt are more separable than those without DiffAtt.

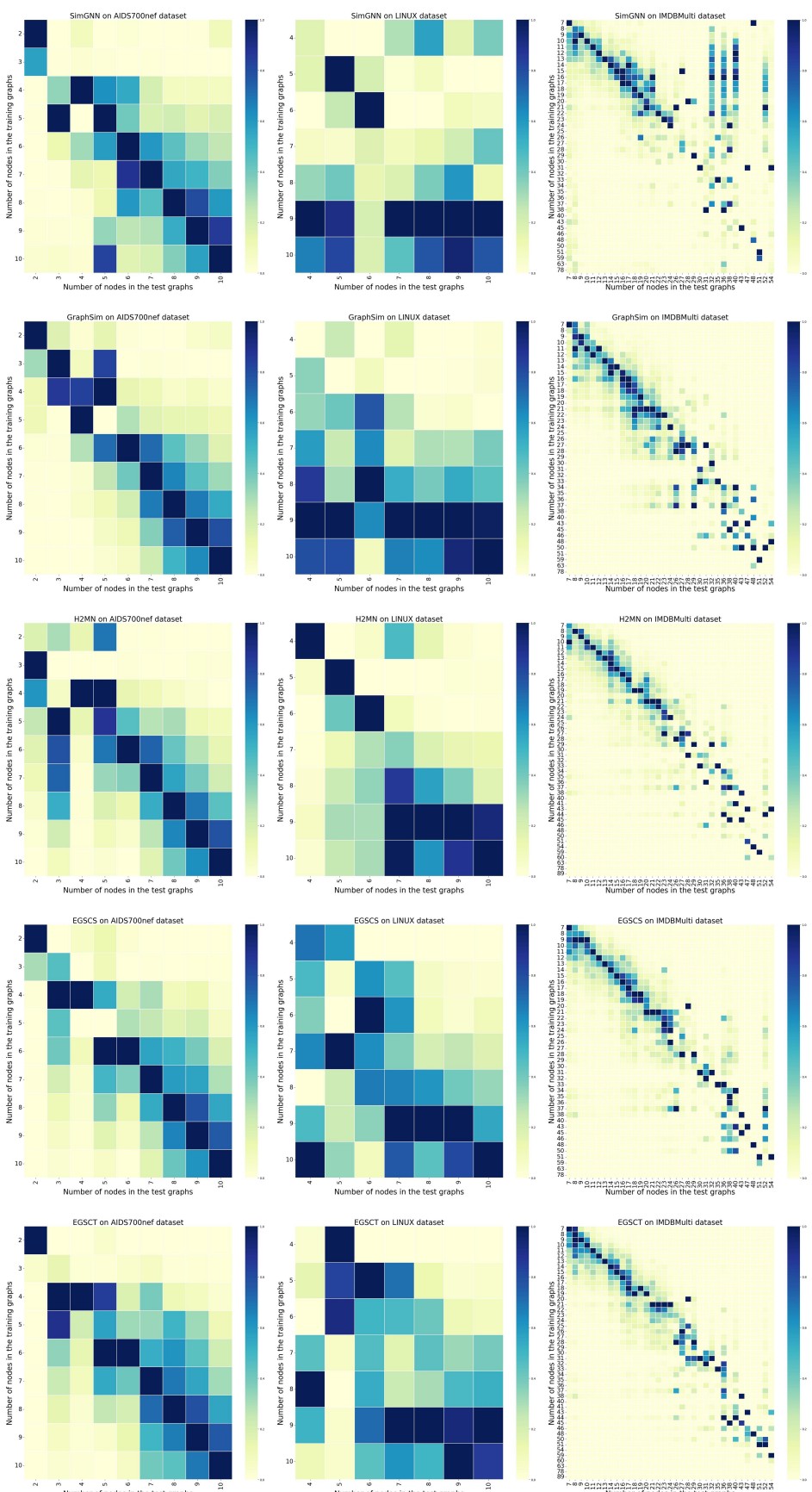

Figure 6: Heatmap of the normalized MSE for different graph similarity models with different number of nodes.

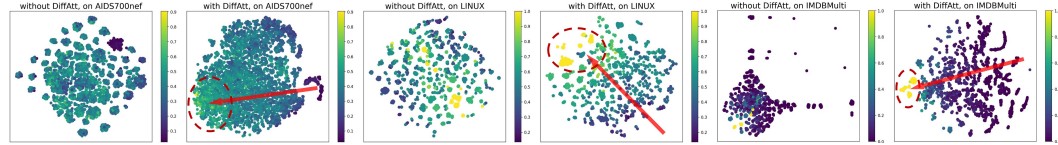

Figure 7: t-SNE visualisation of the joint embeddings generated by models using DiffAtt and those not using DiffAtt. The colours represent the similarity ground truth of the joint embeddings. It is clear that the joint embeddings with DiffAtt are more separable than those without DiffAtt, for example, the similarity scores of the joint embeddings along the arrows gradually increase, and the joint embeddings with high similarity are concentrated in the elliptical region.

Table 7: Results of MSE for different models on test data with small difference in the number of nodes on the AIDS700nef dataset.

| $SizeDiff$ | H2MN | EGSCS | GraphSim | EGSCT | $N^2$AGim |
|---|---|---|---|---|---|
| $[0.000, 0.075)$ | 2.880 | 2.265 | 2.485 | 2.272 | **1.661** |
| $[0.075, 0.150)$ | 2.059 | 1.611 | 1.869 | 1.694 | **1.226** |
| $[0.150, 0.225)$ | 1.442 | 1.260 | 1.555 | 1.428 | **0.898** |
| $[0.225, 0.300)$ | 1.545 | 1.615 | 2.022 | 2.182 | **0.894** |
| $[0.300, 0.375)$ | 0.916 | 0.901 | 0.917 | 1.443 | **0.573** |

## G  ADDITIONAL EXPERIMENTS AND RESULTS

Here we provide more ablation experiments about $N^2$AGim and $GSL^2$ in the same training framework as Section 5.

### G.1  RESULTS ON $N^2$AGIM'S EFFECTIVE IMPROVEMENT OF $N^2$AI

The Fig5 visually demonstrates that $N^2$AGim achieves better performance than the other models when the number of nodes is similar, showing that $N^2$AGim can address $N^2$AI effectively. Here we provide more detailed numerical comparison results in Table 7, 8 and 9. Note that the graphs in AIDS700nef and LINUX all have a relatively small number of nodes, and a 7.5% difference in node number can be seen as a difference of one node. The results demonstrate that $N^2$AGim achieves better performance than the other models at all levels of $SizeDiff$, especially about 26.7%(1.661 vs 2.265), 19.4%(0.191 vs 0.237) and 35.4%(1.103 vs 1.707) better than the second better performance when the difference in number of nodes is less than 7.5% on the three datasets, respectively. This is a strong evidence that $N^2$AGim practically improves $N^2$AI.

### G.2  MORE ABLATION EXPERIMENTS ON $N^2$AGIM

**We experimented with different settings of the backbone of $N^2$AGim on the AIDS700nef dataset.** First we experimented with different types of GNNs and whether to use residual connections and FFNs to enhance node embeddings, and the results are shown in Table 10. It is shown that all GNNs show a significant improvement with the enhancement, especially in the MSE metrics about 13.4%(1.375 vs 1.191), 2.5%(1.254 vs 1.223) and 12.0%(1.330 vs 1.170), respectively.

Table 8: Results of MSE for different models on test data with small differences in the number of nodes on the LINUX dataset.

| $SizeDiff$ | H2MN | EGSCS | GraphSim | EGSCT | $N^2$AGim |
|---|---|---|---|---|---|
| $[0.000, 0.075)$ | 0.709 | 0.474 | 0.258 | 0.237 | **0.191** |
| $[0.075, 0.150)$ | 0.544 | 0.348 | 0.275 | 0.266 | **0.066** |
| $[0.150, 0.225)$ | 0.317 | 0.241 | 0.184 | 0.254 | **0.035** |
| $[0.225, 0.300)$ | 0.114 | 0.292 | 0.138 | 0.138 | **0.021** |
| $[0.300, 0.375)$ | 0.205 | 0.170 | 0.118 | 0.163 | **0.026** |

Table 9: Results of MSE for different models on test data with small differences in the number of nodes on the IMDBMulti dataset.

| $SizeDiff$ | H2MN | EGSCS | GraphSim | EGSCT | N$^2$AGim |
|---|---|---|---|---|---|
| $[0.000, 0.075)$ | 1.737 | 1.707 | 2.397 | 1.748 | **1.103** |
| $[0.075, 0.150)$ | 0.900 | 0.887 | 1.889 | 0.861 | **0.552** |
| $[0.150, 0.225)$ | 0.630 | 0.603 | 1.478 | 0.628 | **0.421** |
| $[0.225, 0.300)$ | 0.609 | 0.547 | 1.040 | 0.537 | **0.431** |
| $[0.300, 0.375)$ | 0.296 | 0.205 | 2.603 | 0.202 | **0.152** |

Table 10: Experimental results on N$^2$AGim with different GNNs and whether to use residual connections and FFNs on the AIDS700nef dataset.

| GNN | Res&FFN | MSE $\downarrow$ | $\rho \uparrow$ | $\tau \uparrow$ | P@10 $\uparrow$ | P@20 $\uparrow$ |
|---|---|---|---|---|---|---|
| GCN | ✗ | 1.375 | 0.898 | 0.756 | 0.615 | 0.686 |
| GCN | ✔ | **1.191** | **0.912** | **0.778** | **0.668** | **0.738** |
| SAGE | ✗ | 1.254 | **0.914** | **0.780** | 0.652 | 0.721 |
| SAGE | ✔ | **1.223** | 0.911 | 0.778 | **0.680** | **0.726** |
| GIN | ✗ | 1.330 | 0.904 | 0.766 | 0.614 | 0.697 |
| GIN | ✔ | **1.170** | **0.916** | **0.783** | **0.672** | **0.736** |

Compared to other GNNs, GIN achieved better results in most cases, especially better on MSE about 1.8%(1.170 vs 1.191) and 4.3%(1.170 vs 1.223), making it more suitable as a backbone for N2AGim. We further experimented with the performance of N$^2$AGim using different numbers of GIN layers, which is shown in Table 11 and found that the number of layers had little effect on performance, so we chose to use 3 layers of GIN.

### G.3 EXPERIMENTAL RESULTS FOR DIFFERENT NUMBERS OF LANDMARKS IN GSL$^2$.

**We experimented with selecting different numbers of landmarks on the performance of GSL$^2$, and the results are shown in Table 12, 13, and 14.** From the experimental results, we can find that increasing the value of $M$ can improve the accuracy of GSL$^2$, but it also increases the inference time. Considering the balance of inference speed and accuracy, we finally chose $M$ as 60, 30, and 70 for three datasets, respectively.

### G.4 EXPERIMENTAL RESULTS FOR GSL$^2$ WITH DIFFERENT RANDOM SELECTED LANDMARKS.

**We next provide experimental results of GSL$^2$ under different random seeds to test the sensitivity of GSL$^2$ to the selected landmarks, which is shown in Table 15, 16, and 17, respectively.** From these results, we can see that the selection of different landmarks affects the performance of GSL$^2$, but the effect is not significant, which shows the robustness of our GSL$^2$.

Table 11: Experimental results on N$^2$AGim with different number of GIN layers on the AIDS700nef dataset.

| Layers | MSE $\downarrow$ | $\rho \uparrow$ | $\tau \uparrow$ | P@10 $\uparrow$ | P@20 $\uparrow$ |
|---|---|---|---|---|---|
| 3 | 1.170 | 0.916 | 0.783 | 0.672 | 0.736 |
| 4 | 1.199 | 0.917 | 0.785 | 0.673 | 0.723 |
| 5 | 1.174 | 0.913 | 0.781 | 0.674 | 0.736 |

Table 12: Experimental results for different numbers of landmarks selected on the AIDS700nef dataset. $M$ denotes the number of landmarks. Considering the balance of inference speed and accuracy, we finally chose $M$ as 60.

| $M$ | $GSL^2$-R (s) | $GSL^2$-F (s) | MSE $\downarrow$ | $\rho \uparrow$ | $\tau \uparrow$ | P@10 $\uparrow$ | P@20 $\uparrow$ |
|---|---|---|---|---|---|---|---|
| 10 | 2.618 | 0.676 | 2.334 | 0.840 | 0.684 | 0.434 | 0.536 |
| 20 | 3.157 | 0.805 | 1.699 | 0.884 | 0.739 | 0.544 | 0.635 |
| 30 | 3.134 | 0.738 | 1.559 | 0.896 | 0.755 | 0.577 | 0.668 |
| 40 | 3.389 | 0.696 | 1.563 | 0.897 | 0.756 | 0.576 | 0.670 |
| 50 | 3.537 | 0.697 | 1.481 | 0.903 | 0.764 | 0.579 | 0.678 |
| 60 | 3.874 | 0.718 | 1.470 | 0.905 | 0.767 | 0.604 | 0.688 |
| 70 | 4.207 | 0.764 | 1.455 | 0.902 | 0.763 | 0.583 | 0.682 |
| 80 | 4.495 | 0.737 | 1.621 | 0.895 | 0.753 | 0.566 | 0.655 |
| 90 | 4.580 | 0.740 | 1.450 | 0.903 | 0.763 | 0.588 | 0.680 |
| 100 | 4.900 | 0.785 | 1.765 | 0.890 | 0.746 | 0.533 | 0.636 |
| 110 | 4.974 | 0.767 | 1.675 | 0.893 | 0.749 | 0.554 | 0.647 |
| 120 | 5.346 | 0.749 | 1.557 | 0.898 | 0.757 | 0.574 | 0.663 |

Table 13: Experimental results for different numbers of landmarks selected on the LINUX dataset. Considering the balance of inference speed and accuracy, we finally chose $M$ as 30.

| $M$ | $GSL^2$-R (s) | $GSL^2$-F (s) | MSE $\downarrow$ | $\rho \uparrow$ | $\tau \uparrow$ | P@10 $\uparrow$ | P@20 $\uparrow$ |
|---|---|---|---|---|---|---|---|
| 10 | 4.254 | 1.015 | 0.696 | 0.981 | 0.920 | 0.944 | 0.939 |
| 20 | 4.283 | 1.035 | 0.382 | 0.987 | 0.944 | 0.975 | 0.951 |
| 30 | 5.007 | 1.159 | 0.074 | 0.994 | 0.964 | 0.995 | 0.991 |
| 40 | 5.235 | 1.132 | 0.080 | 0.994 | 0.967 | 0.990 | 0.988 |
| 50 | 5.467 | 1.052 | 0.084 | 0.994 | 0.967 | 0.990 | 0.993 |
| 60 | 5.753 | 1.017 | 0.081 | 0.994 | 0.967 | 0.991 | 0.993 |
| 70 | 6.018 | 1.040 | 0.088 | 0.993 | 0.960 | 0.987 | 0.989 |
| 80 | 6.634 | 1.039 | 0.209 | 0.991 | 0.957 | 0.973 | 0.960 |
| 90 | 6.763 | 0.970 | 0.077 | 0.994 | 0.964 | 0.989 | 0.989 |
| 100 | 7.703 | 1.115 | 0.130 | 0.993 | 0.962 | 0.982 | 0.976 |
| 110 | 7.752 | 1.111 | 0.212 | 0.990 | 0.947 | 0.974 | 0.960 |
| 120 | 8.268 | 1.111 | 0.087 | 0.994 | 0.966 | 0.993 | 0.994 |

### G.5 EXPERIMENTAL RESULTS ON THE INFERENCE SPEED OF $GSL^2$ BASED ON OTHER MODELS.

**We provide inference speed of $GSL^2$ based on the other graph similarity models and the results are shown in Table 18.** It is clear that $GSL^2$-F speeds up SimGNN by 7.7, 8.5, 73 times on three datasets, respectively, and speeds up GraphSim 6.1, 10.2 and 59 times, respectively.

### G.6 EXPERIMENTAL RESULTS ON $GSL^2$ WITHOUT USING MLPS.

**We provide experiments directly using $\min_i\{\tilde{u}_{G_{1_i}} + \tilde{u}_{G_{2_i}}\}$ and the results are shown in Table 19.** Obviously, due to the limited number of landmarks, and the noise in the generated $u_G$, direct use of $\min_i\{\tilde{u}_{G_{1_i}} + \tilde{u}_{G_{2_i}}\}$ is very ineffective.

### G.7 EXPERIMENTAL RESULTS ON $GSL^2$ WITH DIFFERENT REGRESSION ALGORITHMS.

**We provide experiments using different regression algorithms in $GSL^2$ and the results are shown in the Tab 20 21, 22, respectively**. The experimental setup is the same as in Section 5, and we use the default parameters in Pycaret (Ali, 2020) to train each model. In practice, the parameters of the learning algorithm can be adjusted to obtain better results. It can be seen that the decision tree based regression algorithms achieve good performance in this noisy embeddings. This also illustrates the strong expressive ability of our generated embeddings.

Table 14: Experimental results for different numbers of landmarks selected on the IMDBMulti dataset. Considering the balance of inference speed and accuracy, we finally chose $M$ as 70.

| $M$ | GSL$^2$-R (s) | GSL$^2$-F (s) | MSE $\downarrow$ | $\rho \uparrow$ | $\tau \uparrow$ | P@10 $\uparrow$ | P@20 $\uparrow$ |
|---|---|---|---|---|---|---|---|
| 10 | 6.525 | 1.513 | 0.703 | 0.964 | 0.898 | 0.843 | 0.862 |
| 20 | 7.773 | 1.458 | 0.687 | 0.974 | 0.929 | 0.854 | 0.872 |
| 30 | 8.916 | 1.470 | 0.589 | 0.968 | 0.915 | 0.860 | 0.882 |
| 40 | 10.902 | 1.714 | 0.640 | 0.967 | 0.912 | 0.832 | 0.858 |
| 50 | 12.312 | 1.786 | 0.536 | 0.969 | 0.918 | 0.852 | 0.890 |
| 60 | 13.142 | 1.524 | 0.582 | 0.972 | 0.920 | 0.856 | 0.871 |
| 70 | 15.792 | 1.824 | 0.510 | 0.971 | 0.916 | 0.869 | 0.887 |
| 80 | 17.720 | 1.912 | 0.499 | 0.974 | 0.932 | 0.868 | 0.878 |
| 90 | 18.897 | 1.774 | 0.502 | 0.973 | 0.927 | 0.855 | 0.888 |
| 100 | 20.914 | 1.806 | 0.508 | 0.971 | 0.924 | 0.863 | 0.890 |
| 110 | 24.730 | 1.804 | 0.524 | 0.969 | 0.911 | 0.864 | 0.882 |
| 120 | 26.778 | 1.830 | 0.520 | 0.970 | 0.919 | 0.865 | 0.887 |

Table 15: Experimental results on GSL$^2$ using different random seeds on the AIDS700nef dataset.

| Seed | MSE $\downarrow$ | $\rho \uparrow$ | $\tau \uparrow$ | P@10 $\uparrow$ | P@20 $\uparrow$ |
|---|---|---|---|---|---|
| 2 | 1.470 | 0.905 | 0.767 | 0.604 | 0.688 |
| 3 | 1.499 | 0.904 | 0.765 | 0.599 | 0.681 |
| 4 | 1.509 | 0.904 | 0.765 | 0.591 | 0.682 |
| 5 | 1.477 | 0.902 | 0.763 | 0.574 | 0.667 |
| 2233 | 1.577 | 0.896 | 0.755 | 0.568 | 0.665 |
| std | 0.042 | 0.004 | 0.005 | 0.016 | 0.010 |

## H    LIMITATIONS AND FUTURE WORKS

GSL$^2$ represents each graph as the GED values between it and the landmarks, and learns on these embeddings. However, this restricted number of landmarks and the embeddings with noise limit the performance of GSL$^2$. Moreover, the different randomly chosen landmarks can have some impact on the performance of the GSL$^2$, which requires a better landmark selection strategy to be proposed. We leave these issues for the future works. Besides, this paper also discover the N$^2$AI, a common problem in graph similarity learning, which could inspire the future works.

Table 16: Experimental results on GSL$^2$ using different random seeds on the LINUX dataset.

| Seed | MSE $\downarrow$ | $\rho \uparrow$ | $\tau \uparrow$ | P@10 $\uparrow$ | P@20 $\uparrow$ |
|---|---|---|---|---|---|
| 2 | 0.074 | 0.994 | 0.964 | 0.995 | 0.991 |
| 3 | 0.083 | 0.993 | 0.948 | 0.987 | 0.987 |
| 4 | 0.098 | 0.992 | 0.947 | 0.983 | 0.982 |
| 5 | 0.090 | 0.992 | 0.947 | 0.992 | 0.990 |
| 2233 | 0.084 | 0.993 | 0.948 | 0.984 | 0.991 |
| std | 0.009 | 0.001 | 0.008 | 0.005 | 0.004 |

Table 17: Experimental results on GSL$^2$ using different random seeds on the IMDBMulti dataset.

| Seed | MSE $\downarrow$ | $\rho \uparrow$ | $\tau \uparrow$ | P@10 $\uparrow$ | P@20 $\uparrow$ |
|---|---|---|---|---|---|
| 2 | 0.510 | 0.971 | 0.916 | 0.869 | 0.887 |
| 3 | 0.532 | 0.965 | 0.890 | 0.852 | 0.883 |
| 4 | 0.559 | 0.970 | 0.902 | 0.852 | 0.881 |
| 5 | 0.553 | 0.970 | 0.902 | 0.866 | 0.874 |
| 2233 | 0.562 | 0.968 | 0.898 | 0.863 | 0.881 |
| std | 0.022 | 0.002 | 0.009 | 0.008 | 0.005 |

Table 18: Experimental results on how faster GSL$^2$ can improve other graph similarity models.

| | SimGNN | | | GraphSim | | | N$^2$AGim | | |
|---|---|---|---|---|---|---|---|---|---|
| | AIDS | LINUX | IMDB | AIDS | LINUX | IMDB | AIDS | LINUX | IMDB |
| Original | 5.106 | 8.582 | 122.939 | 4.383 | 9.12 | 114.676 | 9.245 | 13.163 | 87.032 |
| GSL$^2$-R | 3.019 | 4.171 | 11.64 | 4.353 | 5.19 | 15.428 | 3.874 | 5.007 | 15.792 |
| GSL$^2$-F | 0.665 | 1.013 | 1.684 | 0.713 | 0.896 | 1.946 | 0.718 | 1.159 | 1.824 |

Table 19: Experimental results on direct use of $\min_i\{\tilde{\boldsymbol{u}}_{G_{1i}} + \tilde{\boldsymbol{u}}_{G_{2i}}\}$.

| Datasets | MSE $\downarrow$ | $\rho \uparrow$ | $\tau \uparrow$ | P@10 $\uparrow$ | P@20 $\uparrow$ |
|---|---|---|---|---|---|
| AIDS700nef | 13.283 | 0.683 | 0.527 | 0.246 | 0.299 |
| LINUX | 3.133 | 0.961 | 0.913 | 0.767 | 0.811 |
| IMDBMulti | 1.802 | 0.955 | 0.897 | 0.721 | 0.762 |

Table 20: Experimental results on GSL$^2$ using different regression algorithms on the AIDS700nef dataset.

| Model | MSE ↓ | $\rho$ ↑ | $\tau$ ↑ | P@10 ↑ | P@20 ↑ |
|---|---|---|---|---|---|
| Extra Trees | 1.705 | 0.888 | 0.751 | 0.541 | 0.637 |
| CatBoost | 1.645 | 0.890 | 0.746 | 0.559 | 0.656 |
| Random Forest | 1.963 | 0.874 | 0.729 | 0.475 | 0.583 |
| KNeighbors | 1.909 | 0.872 | 0.732 | 0.561 | 0.657 |
| XGBoost | 1.870 | 0.874 | 0.724 | 0.524 | 0.618 |
| LightGBM | 2.317 | 0.849 | 0.694 | 0.456 | 0.567 |
| Gradient Boosting | 3.305 | 0.796 | 0.635 | 0.291 | 0.434 |
| Decision Tree | 4.854 | 0.743 | 0.609 | 0.382 | 0.479 |
| Bayesian Ridge | 6.108 | 0.504 | 0.383 | 0.066 | 0.139 |
| Linear Regression | 6.108 | 0.504 | 0.384 | 0.072 | 0.138 |
| Ridge | 6.108 | 0.504 | 0.384 | 0.071 | 0.138 |
| Orthogonal Matching Pursuit | 6.291 | 0.490 | 0.371 | 0.059 | 0.124 |
| Huber | 6.218 | 0.501 | 0.382 | 0.071 | 0.135 |
| ElasticNet | 6.470 | 0.483 | 0.372 | 0.077 | 0.143 |
| Passive Aggressive Regressor | 6.603 | 0.477 | 0.367 | 0.076 | 0.133 |
| Lasso | 7.325 | 0.432 | 0.333 | 0.076 | 0.143 |
| AdaBoost | 8.754 | 0.383 | 0.304 | 0.314 | 0.347 |
| LassoLars | 10.776 | 0.226 | 0.186 | 0.466 | 0.490 |
| MLPs | 1.470 | 0.905 | 0.767 | 0.604 | 0.688 |

Table 21: Experimental results on GSL$^2$ using different regression algorithms on the LINUX dataset.

| Model | MSE ↓ | $\rho$ ↑ | $\tau$ ↑ | P@10 ↑ | P@20 ↑ |
|---|---|---|---|---|---|
| Extra Trees | 0.078 | 0.999 | 0.996 | 0.987 | 0.991 |
| Random Forest | 0.103 | 0.998 | 0.994 | 0.981 | 0.981 |
| XGBoost | 0.175 | 0.992 | 0.964 | 0.978 | 0.967 |
| KNeighbors | 0.141 | 0.998 | 0.995 | 0.986 | 0.984 |
| CatBoost | 0.148 | 0.993 | 0.960 | 0.964 | 0.977 |
| Decision Tree | 0.222 | 0.997 | 0.994 | 0.987 | 0.988 |
| LightGBM | 0.980 | 0.979 | 0.921 | 0.951 | 0.949 |
| Gradient Boosting | 4.450 | 0.944 | 0.847 | 0.743 | 0.819 |
| AdaBoost | 24.366 | 0.591 | 0.472 | 0.298 | 0.300 |
| ElasticNet | 28.596 | 0.365 | 0.288 | 0.090 | 0.145 |
| Orthogonal Matching Pursuit | 30.056 | 0.319 | 0.265 | 0.045 | 0.045 |
| Lasso | 31.502 | 0.317 | 0.265 | 0.090 | 0.145 |
| Huber | 28.490 | 0.395 | 0.310 | 0.090 | 0.145 |
| Bayesian Ridge | 26.908 | 0.386 | 0.299 | 0.090 | 0.145 |
| Ridge | 26.908 | 0.388 | 0.305 | 0.090 | 0.145 |
| Linear Regression | 26.908 | 0.388 | 0.305 | 0.090 | 0.145 |
| MLPs | 0.074 | 0.994 | 0.964 | 0.995 | 0.991 |

Table 22: Experimental results on GSL$^2$ using different regression algorithms on the IMDBMulti dataset.

| Model | MSE ↓ | $\rho$ ↑ | $\tau$ ↑ | P@10 ↑ | P@20 ↑ |
|---|---|---|---|---|---|
| Extra Trees | 0.585 | 0.971 | 0.935 | 0.875 | 0.890 |
| Random Forest | 0.776 | 0.968 | 0.931 | 0.862 | 0.883 |
| XGBoost | 2.901 | 0.951 | 0.884 | 0.754 | 0.803 |
| KNeighbors | 0.837 | 0.961 | 0.924 | 0.876 | 0.899 |
| LightGBM | 13.573 | 0.925 | 0.837 | 0.728 | 0.758 |
| Decision Tree | 1.275 | 0.954 | 0.914 | 0.841 | 0.875 |
| Gradient Boosting | 48.646 | 0.825 | 0.700 | 0.206 | 0.320 |
| Bayesian Ridge | 75.886 | 0.333 | 0.250 | 0.015 | 0.034 |
| Linear Regression | 75.887 | 0.333 | 0.251 | 0.015 | 0.034 |
| ElasticNet | 75.690 | 0.330 | 0.252 | 0.013 | 0.033 |
| Lasso | 75.779 | 0.330 | 0.252 | 0.013 | 0.033 |
| Ridge | 75.959 | 0.336 | 0.254 | 0.015 | 0.036 |
| Least Angle Regression | 76.013 | 0.333 | 0.247 | 0.015 | 0.036 |
| CatBoost | 2.100 | 0.957 | 0.896 | 0.817 | 0.825 |
| Huber | 90.815 | 0.351 | 0.265 | 0.025 | 0.032 |
| Orthogonal Matching Pursuit | 174.995 | 0.236 | 0.178 | 0.022 | 0.045 |
| MLPs | 0.510 | 0.971 | 0.916 | 0.869 | 0.887 |

