# OpenReview forum: "Node Number Awareness Representation for Graph Similarity Learning"
_ICLR.cc/2023/Conference — Submitted to ICLR 2023_

### Official Review · Reviewer_ngMc · 2022-10-20

**Confidence:** 3
**Correctness:** 3
**Technical Novelty And Significance:** 3
**Empirical Novelty And Significance:** 3
**Recommendation:** 3

**Clarity, Quality, Novelty And Reproducibility:**

Clarity: Poor. This paper aims to address two kinds of issues, but do not give any connections between these two issues.

Quality: Poor. The paper has technical flaws. For example, the proof of the main theorem is incorrect or the experimental evaluation is flawed and fails to adequately support the main claims.

Novelty: Fair. The paper contributes some new ideas or represents incremental advances.

Reproducibility: Excellent. Key resource are available and key details are comprehensively described such that competent researchers will be able to easily reproduce the main results.


**Strength And Weaknesses:**

Strength:
1. They found the existing graph similarity models have a relatively large error in predicting the actual similarity of two graphs with similar number of nodes, because of the global pooling function, which is interesting.
2. They propose A novel GNN-based graph similarity model, named N2AGim.
3. They propose the GSL2 to speed up the inference of graph similarity models.

Weaknesses:
1. The writing level of the paper needs to be improved, the expression is not organized, and there are problems with the writing structure.
2. Lack of the survey for related work.
3. Why we need to address to accelerate similarity computation by GSL2, and what is the connection of this strategy to N2AI issue?



**Summary Of The Paper:**

This paper aims to solve Node Number Awareness Issue (N2AI) and accelerate the inference speed in graph similarity learning.
They argue that existing graph learning methods tend to map graphs with similar number of nodes to similar embedding distributions, reducing the separability of their embeddings. They also use (GSL2) to accelerate similarity computation. They conduct extensive experiments to demonstrate their effectiveness.


**Summary Of The Review:**

The logic of the paper is not clear, the focus is not highlighted, and it is difficult to understand their innovation points and contributions.
Please refer to "weakness" above.

---

> ### Author Response · Authors · 2022-11-12
> **Response to Reviewer ngMc (2/2)**
>
> > Q4 : This paper aims to address two kinds of issues, but do not give any connections between these two issues.
>
> Please see Q3.1 for a better understanding of the connection between our two proposed models.
>
> > Q5 : The paper has technical flaws. For example, the proof of the main theorem is incorrect or the experimental evaluation is flawed and fails to adequately support the main claims.
>
> Thank you for this question. We offer here a summary of our paper, which we hope will help you to understand this paper more clearly.
>
> 1. This work aims to address two important issues in the graph similarity computation, the first one is the **Node Number Awareness Issue (N2AI)**, and the second one is how to **accelerate the inference speed of graph similarity computation**.
> 2. We found that existing Graph Neural Network based graph similarity models have a large error in predicting the similarity scores of two graphs with similar number of nodes, and it is shown in Fig1. Our analysis in Sec3 shows that this is because of the global pooling function in graph neural networks that maps graphs with similar number of nodes to similar embedding distributions, reducing the separability of their embeddings, which we refer to as the **N2AI**.
> 3. In Sec4.1, we leverage our proposed Different Attention (DiffAtt) to construct Node Number Awareness Graph Similarity Model (N2AGim) to address the N2AI. To verify this, the following experiments were conducted:
> 	1. In Table1 of Sec5.1, *we report the results of ablation experiments with and without DiffAtt on all four pooling methods.* Table 1 demonstrates that the DiffAtt effectively improves 44 metrics out of 48 metrics of the four global pooling methods. **This shows that DiffAtt has powerful generalization to effectively address the N2AI of all pooling methods.**
> 	2. In Table 2 of Sec5.1, *we report experiments comparing DiffAtt with other attention mechanisms under the same structure*, and DiffAtt achieves the best results on all metrics. This illustrates that **DiffAtt is more effective than other attention mechanisms.**
> 	3. In Sec5.2, *we compared N2AGim with other mainstream graph similarity models in the graph similarity task*, and the results in Table4 demonstrate that **N2AGim achieves the state-of-the-art performance.**
> 	4. Moreover, *we show the performance of different models with small differences in the number of graph nodes in Fig5 and Appendix G.1 of the new submitted paper.* These results demonstrate that **N2AGim performs better when the number of graph nodes is similar, which is a strong indication that N2AGim solves N2AI effectively.**
> 4. In Section 4.2, we propose the GSL2 framework to accelerate graph similarity inference based on Theorem 1.  The following experiments were conducted:
> 	1. Based on Theorem 1, *we argue that the accuracy of the generated landmarks has an impact on the performance of GSL2.* In Table 3 of Sec5.1, we show that GSL2 using the more accurate N2AGim achieves better results than SimGNN-based and GraphSim-based. **This is a strong evidence that addressing N2AI can improve the performance of GSL2 and therefore N2AGim is the most suitable GED calculator for GSL2.**
> 	2. *We provide the inference times required for each model on the three datasets in Sec5.3.* The results show that GSL2 comes out to be 12.9, 11.3 and 47.7 times faster than N2AGim on the three datasets, respectively, and compared to EGSCS, GSL2 is 1.36, 1.22 and 1.24 times faster, respectively. This illustrates that **GSL2 can effectively improve the speed of graph similarity inference. **
> 	3. *We also have experimented with the accuracy of GSL2 when using different numbers of landmark in Appendix F (G.3 of the new submission).* It is concluded that **increasing the number of landmarks improves the performance of GSL2**, and we finally chose 60, 30, and 70 as the number of GSL2 landmarks on the three datasets.
> 	4. *We also experimented with the effect of using different randomly selected landmarks* on GSL2 in Appendix F (G.4 of the new submission) and showed that the effect was not significant, which illustrates **the robustness of GSL2.**
>
> In addition, we would be grateful if you could specifically point out errors in our proof or problems in our experimental evaluation. We trust that your further comments will improve the quality of the paper.
>
> [1]Gilmer J, et al. Neural message passing for quantum chemistry
> [2]Leskovec, et al. How powerful are graph neural networks
> [3]Li Y, et al. Graph matching networks for learning the similarity of graph structured objects
> [4]Bai Y, et al. Simgnn: A neural network approach to fast graph similarity computation
> [5]Bai Y, et al. Learning-based efficient graph similarity computation via multi-scale convolutional set matching
> [6]Qin C, et al. Slow Learning and Fast Inference: Efficient Graph Similarity Computation via Knowledge Distillation
>
> _Please feel free to let us know if you have further questions._
>
> Best, Authors

---

> ### Author Response · Authors · 2022-11-12
> **Response to Reviewer ngMc (1/2)**
>
> We thank the reviewer for the constructive comments.
>
> > Q1 : The writing level of the paper needs to be improved, the expression is not organized, and there are problems with the writing structure.
>
> Thank you for your suggestions. In fact, we focused on logical clarity and correct expression in our writing and we revised the paper several times before submitting it. We also had several native English speakers who helped us to improve the paper to ensure that it was well written.
>
> > Q2 : Lack of the survey for related work.
>
> Indeed, **we have already provided a survey of related work in Sec2.** Specifically, we provide a survey on Graph Neural Networks in Sec2.1, e.g., an introduction to the MPNN[1] architecture and the GIN[2] architecture. In Sec2.2, we provide an introduction to all the popular GNN-based graph similarity models, including GMN[3], SimGNN[4], GraphSim[5], EGSC[6] , etc.
>
> > Q3 : Why we need to address to accelerate similarity computation by GSL2, and what is the connection of this strategy to N2AI issue?
>
> > Q3.1 : Why we need to address to accelerate similarity computation by GSL2?
>
> The existing solution [6] uses a specially designed knowledge distillation (KD) paradigm to accelerate graph similarity inference. Compared to [6], the advantages of our GSL2 are as follows:
>
> * **[Clear theory support.]** GSL2 has a clear theory which ensures that GSL2 can be improved by increasing the accuracy of the GED calculator and by selecting better landmarks. This theory also ensures that different graph similarity models as well as regression algorithms can be selected depending on the practical problem, which greatly expands the scenarios in which the GSL2 framework can be used.
> * **[Plug and Play.]** GSL2 is plug and play and can be applied to any GED calculator, whereas the distillation process of [6] is not a separate module that can be plugged and played.
> * **[Better performance.]** GSL2 outperforms the student model in [6] on 10 of the 12 metrics(see Table 4), and is particularly better on MSE for each of the three datasets 4.9%(1.470 vs 1.546), 74.7%(0.074 vs 0.293) and 12.2%(0.510 vs 0.581), respectively. In addition, compared to the student model in [6], GSL2 is 1.36, 1.22 and 1.24 times faster on three datasets(see Table 5), respectively.
>
> > Q3.2 : What is the connection of this strategy to N2AI issue?
>
> Thank you for your question. As we have analysed in Sec4.2, the performance of GSL2 relies on the high accuracy of the GED calculator. The N2AI is the key to improving the accuracy of the GED calculator, so **the performance of GSL2 can be improved by effectively addressing the N2AI.** To verify this, we experimented with GSL2 for different accuracy graph similarity models in Sec5.1, and the results in Table3 illustrate that, compared to graph similarity models that do not focus on addressing N2AI, N2AGim-based GSL2 achieves the best results in 11 out of 12 metrics, especially on the MSE of the LINUX dataset than the SimGNN-based and GraphSim-based ones by 81.6% (0.074 vs 0.402) and 64.8% (0.074 vs 0.210). This is a strong evidence that addressing N2AI can improve the performance of GSL2 and therefore N2AGim is the most suitable GED calculator for GSL2. **In summary, addressing N2AI helps to improve the accuracy of GSL2, which in turn helps to improve the performance of graph similarity inference.**

---

> ### Author Response · Authors · 2022-11-18
> **Looking forward to your post-rebuttal feedback**
>
> Thanks again for your insightful suggestions and comments. As the deadline for discussion is approaching, we are glad to provide any additional clarifications that you may need, and here we would like to highlight the contributions of this paper :
> * We found and analysed N2AI in the graph similarity model, and our conclusion is that **_all_ graph pooling modules map two graphs with a similar number of nodes to similar embedding distribution, thus reducing the separability between embeddings.**
> * We propose the DiffAtt module and the N2AGim model to address N2AI. Experimental results show that:
> 	1. DiffAtt can substantially improve performance on all four pooling modules.
> 	2. DiffAtt is superior to other attention mechanisms in terms of effectiveness.
> 	3. N2AGim achieves the **state-of-the-art performance** in graph similarity learning.
> 	4. N2AGim performs better than any other model on graphs with similar number of nodes, which shows that **N2AGim can effectively address N2AI**.
> * We propose the GSL2 framework to accelerate graph similarity inference. Experimental results show that:
> 	1. The accuracy of GSL2 can be improved by using a graph similarity model with high accuracy. **Therefore, using a model that can effectively solve N2AI to achieve higher accuracy, i.e. N2AGim, can effectively improve the accuracy of GSL2.**
> 	2. N2AGim-based GSL2 can be **up to 12.9, 11.3 and 47.7 times faster** than N2AGim.
> 	3. Compared to other models, GSL2 achieves the **state-of-the-art performance** in terms of inference speed.
>
> _We hope our responses convincingly address your concerns. Please do not hesitate to contact us if there are other clarifications or experiments we can offer_.
>
> _Thank you for your time again!_
>
> Best, Authors

---

> ### Author Response · Authors · 2022-11-23
> **Looking forward to the post-rebuttal feedbacks again!**
>
> It's been more than **10** days since we submitted our response and we still haven't received your feedback.
>
> Please don’t hesitate to let us know if there are any additional clarifications or experiments that we can offer, as we would love to convince you of the merits of the paper. We appreciate your suggestions.
>
> Thanks!
>
> Best, Authors

---

> ### Author Response · Authors · 2022-12-04
> **Looking forward to your post-rebuttal feedbacks again!**
>
> It's been more than **22** days since we submitted our response and we still haven't received your feedback.
>
> Please don’t hesitate to let us know if there are any additional clarifications or experiments that we can offer, as we would love to convince you of the merits of the paper. We appreciate your suggestions.
>
> Thanks!
>
> Best, Authors

---

### Official Review · Reviewer_qS53 · 2022-10-22

**Confidence:** 4
**Correctness:** 3
**Technical Novelty And Significance:** 2
**Empirical Novelty And Significance:** 3
**Recommendation:** 6

**Clarity, Quality, Novelty And Reproducibility:**

The clarity of the paper is good except for some typos and annotation errors.

The quality and novelty of the paper are also good but there is a concern about the experiment result. Is it fair to compare with those baselines with the reported results?

The author has provided the open source code, which ensures the reproducibility.


**Strength And Weaknesses:**

Strength
- This paper proposes an interesting problem (N^2AI), which means graphs with a similar number of nodes would be mapped to similar embedding distribution.
- The author proposes N^2AGIM to solve the found problem.
- The author proposes GSL^2 to speed up graph similarity learning inference and it’s plug-and-play.
- The experiment results validate the effectiveness of proposed methods.

Weakness
- The motivation of some parts of the model design is unclear, e.g., why using GIN can effectively address N^2AI as mentioned in section 4.1(Multi-scale GIN layers). Have you tried using other GNN backbones? / Why do you use three GIN layers in N^2AGIM? (It seems this should be a hyperparameter.) If that is the case, then a parameter analysis regarding this parameter can be conducted.
- More detailed inference time comparison is encouraged to be added. For example, as a plug-and-play module. How fast can it improve on the top of graph similarity learning models, e.g., SimGNN and GraphSim?
- The experiment result is my primary concern. It seems it is not consistent with the results reported in other papers. For example, in H2MN, they show their MSE is 0.913 for AIDS, and 0.105 for LINUX respectively. Nevertheless, in your work, you show that their results for these two datasets are respectively 1.826 and 0.21. Is there any difference between your experiment setting and their experiment setting?
- It is suggested that annotations errors and typos should be eliminated in the work. In section 3, $F_i$ and $\mathcal F_i$ are both used for the pooling method. In addition, X and $X$ are both used for representing feature matrix. In Theorem 1, $i$ is used to denote which graph for $G_i \in S$, while it is also used to denote $u_{G_{1i} $ (which is unclear what i means here).


**Summary Of The Paper:**

This work found that the existing graph similarity models have a relatively large error in predicting the actual similarity of two graphs with a similar number of nodes. This phenomenon is the so-called Node Number Awareness Issue (N^2AI). This problem is addressed by a novel GNN-based method, called N2AGIM. The authors also propose using GSL^2 to speed up the inference of graph similarity models. GSL^2 can take different graph similarity learning models as backbones, while it reaches the best performance with N^2AGIM. The experiments result demonstrates the superiority of the proposed methods.

**Summary Of The Review:**

This paper proposes an interesting problem (N^2AI), which means graphs with a similar number of nodes would be mapped to similar embedding distribution. The author proposes N^2AGIM to solve the found problem. In addition, the author proposes GSL^2 to speed up graph similarity learning inference and it’s plug-and-play. The experiment results validate the effectiveness of the proposed methods.

However, the motivation of some parts of the model design is unclear, and the experiment result is inconsistent with results reported in other papers.

---

> ### Author Response · Authors · 2022-11-12
> **Response to Reviewer qS53 (2/2)**
>
>
>
> > Q3 : The experiment result is my primary concern. It seems it is not consistent with the results reported in other papers. For example, in H2MN, they show their MSE is 0.913 for AIDS, and 0.105 for LINUX respectively. Nevertheless, in your work, you show that their results for these two datasets are respectively 1.826 and 0.21. Is there any difference between your experiment setting and their experiment setting?
>
> Thank you for your careful reading of our paper! In the first paragraph of section 5.2, we have declared that **we note discrepancies in the MSE reported by various papers. For examples, the MSE reported in SimGNN, GraphSim is $\frac{1}{2 \dot |D|}\sum_{i=1}^{D}(s-\hat{s})^2$, but EGSC reported MSE as $\frac{1}{|D|} \sum_{i=1}^{D}(s-\hat{s})^2$. To provide a consistent comparison, we use the MSE metric of the latter formula.** Evidence of this can be found in the official github code for these models：
> * GMN, SimGNN and GraphSim is evaluated and reported by Bai et al, and Bai acknowledges their use of denominator 2 in the official code issue [L1]. The code for their MSE implementation can be found at [L2] (SimGNN) and [L3] (GraphSim).
> * H2MN also uses the MSE with a denominator of 2, and Please See [L4].
> * EGSC uses an MSE without denominator 2, and Please See [L5].
> However, most open source code for deep learning uses MSE without denominator 2[L6,L7]. In order not to create greater misunderstanding, we decided to use the more extensive MSE indicator without denominator 2 in the paper and to correct all the indicators. Therefore, our experiments are more fair in comparison.
>
> [L1] https://github.com/yunshengb/GraphSim/issues/3#issuecomment-744267413
> [L2] https://github.com/yunshengb/SimGNN/blob/2abc5a367e7ac0a5779d1979cbb510c53a4a15f9/src/metrics.py#L78
> [L3]https://github.com/yunshengb/GraphSim/blob/f73ba796e0d20ee1b1fa0f509f2fcb1df3ac5a28/src/metrics.py#L78
> [L4]https://github.com/cszhangzhen/H2MN/blob/71712acc6f917f399d9f8d3eb1ee31ebbdf9db39/main_regression.py#L156
> [L5]https://github.com/canqin001/Efficient_Graph_Similarity_Computation/blob/6fe3132333b40425983cf8aad435ba00324c67a7/EGSC-T/src/egsc.py#L281
> [L6]https://pytorch.org/docs/stable/generated/torch.nn.MSELoss.html?highlight=mse#torch.nn.MSELoss
> [L7]https://www.tensorflow.org/api_docs/python/tf/keras/losses/MeanSquaredError
>
> > Q4 : It is suggested that annotations errors and typos should be eliminated in the work. In section 3, $F_i$ and $\mathcal{F_i}$ are both used for the pooling method. In addition, X and $X$ are both used for representing feature matrix. In Theorem 1, $i$ is used to denote which graph for $G_i \in \mathcal{S}$, while it is also used to denote $\boldsymbol{u}_{{G_1}_i}$ (which is unclear what i means here).
>
> Thank you for pointing it out. We have corrected the corresponding parts in the resubmission. In addition, to avoid the lack of clarity in Eq3, we re-clarify it in the new submission as:
>
> $$
> GED(G_1,G_2) =  \mathop{\min}_{i} \\{GED(G_1,\hat{G}_i) + GED(G_2,\hat{G}_i) \\},
> $$
>
> where $\hat{G}_i \in \mathcal{S}$. This equation expresses that the $GED(G_1,G_2)$ is equal to $GED(G_1,\hat{G}_i) + GED(G_2,\hat{G}_i)$ if one finds a $\hat{G}_i$ that minimizes $GED(G_1,\hat{G}_i) + GED(G_2,\hat{G}_i)$.
>
> _We sincerely appreciate your comments. Please feel free to let us know if you have further questions._
>
> Best, Authors

---

> > ### Comment · Reviewer_qS53 · 2022-11-17
> > **Feedback**
> >
> > I appreciate the authors' response to my concerns. My major concerns were clarified, and I increased my rating to 6.

---

> ### Author Response · Authors · 2022-11-12
> **Response to Reviewer qS53 (1/2)**
>
> Thank you very much for the careful reading and the insightful comments!
>
> > Q1 : The motivation of some parts of the model design is unclear, e.g., why using GIN can effectively address N2AI as mentioned in section 4.1(Multi-scale GIN layers). Have you tried using other GNN backbones? / Why do you use three GIN layers in N2AGIM? (It seems this should be a hyperparameter.) If that is the case, then a parameter analysis regarding this parameter can be conducted.
>
> Thank you for your suggestion. The GIN is known for having at most as powerful as WL-Test, that is, to distinguish whether two graphs are isomorphic. Thus, we believe that GIN has the theoretical ability to address N2AI, i.e. to distinguish between two graphs with a similar number of nodes. To verify this and to explain the design of N2AGim more clearly, **we have added ablation experiments on N2AGim's backbone in Appendix G.2 of the new resubmitted paper, and the results are shown in Table 10 and Table 11.** The results illustrate that using residual connections and FFNs leads to 13.4%(1.191 vs 1.375), 2.5%(1.223 vs 1.254) and 12.0%(1.170 vs 1.330) improvement in MSE for the three GNN settings, i.e. GCN[1], SAGE[2] and GIN[3], respectively. Compared to GCN and SAGE, GIN achieved better results in most cases, especially better on MSE about 1.8%(1.170 vs 1.191) and 4.3%(1.170 vs 1.223), making it more suitable as the backbone for N2AGim. We have also experimented with different number of GIN layers and the results in Table 11 show that increasing the number of GIN layers has little impact on model performance, so we chose to use 3 layers of GIN.
>
> [1]Kipf T N, Welling M. Semi-supervised classification with graph convolutional networks[J]. arXiv preprint arXiv:1609.02907, 2016.
> [2]Hamilton W, Ying Z, Leskovec J. Inductive representation learning on large graphs[J]. Advances in neural information processing systems, 2017, 30.
> [3]Xu K, Hu W, Leskovec J, et al. How powerful are graph neural networks?[J]. arXiv preprint arXiv:1810.00826, 2018.
>
> > Q2 : More detailed inference time comparison is encouraged to be added. For example, as a plug-and-play module. How fast can it improve on the top of graph similarity learning models, e.g., SimGNN and GraphSim?
>
> That's a very good question! We test the speed of GSL2 based on SimGNN and GraphSim and the results are shown in the table below. The results show that GSL2-F speeds up SimGNN by 7.7, 8.5, 73 times on three datasets, respectively, and speeds up GraphSim 6.1, 10.2 and 59  times on three datasets, respectively. We have also added the results of this experiment to the Appendix G.5  of the new submission paper.
>
> |        |            | SimGNN |           |            | GraphSim |            |
> |--------|------------|--------|-----------|------------|----------|------------|
> |        | AIDS700nef | LINUX  | IMDBMulti | AIDS700nef | LINUX    | IMDBMulti  |
> | Original | 5.106      | 8.582  | 122.939   | 4.383      | 9.12     | 114.676    |
> | GSL2-R   | 3.019      | 4.171  | 11.64     | 4.353      | 5.19     |     15.428       |
> | GSL2-F | 0.665      | 1.013  | 1.684     | 0.713      | 0.896    |    1.946        |

---

### Official Review · Reviewer_w3qo · 2022-10-25

**Confidence:** 2
**Correctness:** 3
**Technical Novelty And Significance:** 3
**Empirical Novelty And Significance:** 3
**Recommendation:** 6

**Clarity, Quality, Novelty And Reproducibility:**

Clarity is good, the paper has a clear logic.

Quality is good, the paper also has theoretical analysis on their method

Novelty is good.

Reproducibility should be good, the code repository provided looks well maintained.

**Strength And Weaknesses:**

Strengths:

This paper has a clear motivation to address the two specific problems in graph similarity computation, therefore the main logic is easy to follow.

Theoretical analysis is provided.

The experimental results are promising.

The paper is well formatted.


Weakness:

1. The inference time, which is one of the two main contributions of this paper, seems not improved significantly compared to the second best method EGSCS-F. Does this means that EGSCS-F is also a highly efficient method and does not suffer from the inference time issue?

2. It is not clear whether the performance is brought about by better performance on the graphs with similar numbers of nodes. It would be better to show the performance of different methods on the graph pairs with similar number of nodes.

**Summary Of The Paper:**

This papertargets two issues in graph similarity computation including the node number awareness issue and the inference speed issue. The authors first analyze and show that the underlying reason of the first issue is the graph pooling. Then, the Different Attention is proposed to get the model aware of the node number. Finally, Graph Similarity Learning with Landmarks is proposed to address the second issue on the computation speed. Experiments are conducted on multiple datasets and compared with multiple baselines.

**Summary Of The Review:**

Overall, this paper has a clear motivation and a reasonable solution. The experiments are overall acceptable. I would recommend acceptance,

---

> ### Author Response · Authors · 2022-11-12
> **Response to Reviewer w3qo**
>
>
> We appreciate the positive and constructive comments from you!
>
> > Q1 : The inference time, which is one of the two main contributions of this paper, seems not improved significantly compared to the second best method EGSCS-F. Does this means that EGSCS-F is also a highly efficient method and does not suffer from the inference time issue?
>
> Thanks for the question. In fact, GSL2 and EGSCS[1] are two different efficient algorithms for accelerated graph similarity inference, and in the offline case (with the suffifix of ’-F’) both simplify the graph similarity inference as $s(G_i,G_j) = MLP(h_i,h_j)$, where $h_i$ is the individal embedings of graph $G_i$. This shows that the difference in time between the two of them will not be significant. The experimental results also illustrate that the speed differences between GSL2-F and EGSCS-F on the three datasets are not significant, i.e. 0.718s and 0.975s on AIDS700nef, 1.159s and 1.414s on LINUX, and 1.824s and 2.256s on the IMDBMulti dataset, respectively.
>
> [1]Qin C, Zhao H, Wang L, et al. Slow Learning and Fast Inference: Efficient Graph Similarity Computation via Knowledge Distillation[J]. Advances in Neural Information Processing Systems, 2021, 34: 14110-14121.
>
> > Q2 :  It is not clear whether the performance is brought about by better performance on the graphs with similar numbers of nodes. It would be better to show the performance of different methods on the graph pairs with similar number of nodes.
>
>
> That's a very good question! Fig1 demonstrates that the accuracy of the GNN-based similarity models decreases as the SizeDiff, the percentage difference in the number of nodes in the two graphs, decreases, which is defined as the N2AI. **To verify whether N2AGim can effectively address N2AI, we visualize the performance of different models on graph pairs with similar numbers of nodes in Fig5 of the resubmitted paper, and provide detailed numerical comparisons in the Appendix G.1.** For example, the comparison on AIDS700nef is shown below :
>
> | SizeDiff | H2MN  | EGSCS | GraphSim | EGSCT | N2AGim  |
> |----------------|-------|-------|----------|-------|---------|
> | \[0.000, 0.075)   | 2.880  | 2.265 | 2.485    | 2.272 | **1.661**   |
> | \[0.075, 0.150)  | 2.059 | 1.611 | 1.869    | 1.694 | **1.226**   |
> | \[0.150, 0.225)  | 1.442 | 1.260  | 1.555    | 1.428 | **0.898**   |
> | \[0.225, 0.300)   | 1.545 | 1.615 | 2.022    | 2.182 | **0.894**   |
> | \[0.300, 0.375)   | 0.916 | 0.901 | 0.917    | 1.443 | **0.573**   |
>
> Note that the graphs in AIDS700nef all have a relatively small number of nodes, and a 7.5% difference in node number can be seen as a difference of approximately one node. The results demonstrate that N2AGim achieves better performance than the other models at all levels of SizeDiff, especially about 26.7% better than the second place EGSCS (1.661 vs 2.265 on MSE) when the difference in number of nodes is less than 7.5%. This is a strong evidence that N2AGim practically improves N2AI. Please see Fig5 and Appendix G.1 in the resubmitted paper for further details.
>
> _We hope that the provided new experiments and additional explanations have convinced you of the merits of our work. Please do not hesitate to contact us if there are other clarifications or experiments we can offer._
>
> Best, Authors

---

> > ### Comment · Reviewer_w3qo · 2022-11-19
> > **post-rebuttal response**
> >
> > Thanks for the explanations from the authors. I have read the responses and choose to keep my rating as acceptance

---

> ### Author Response · Authors · 2022-11-18
> **Looking forward to your post-rebuttal feedback**
>
> Thanks again for your insightful suggestions and comments. As the deadline for discussion is approaching, we are glad to provide any additional clarifications that you may need, and here we would like to highlight the contributions of this paper :
> * We found and analysed N2AI in the graph similarity model, and our conclusion is that **_all_ graph pooling modules map two graphs with a similar number of nodes to similar embedding distribution, thus reducing the separability between embeddings.**
> * We propose the DiffAtt module and the N2AGim model to address N2AI. Experimental results show that:
> 	1. DiffAtt can substantially improve performance on all four pooling modules.
> 	2. DiffAtt is superior to other attention mechanisms in terms of effectiveness.
> 	3. N2AGim achieves the **state-of-the-art performance** in graph similarity learning.
> 	4. N2AGim performs better than any other model on graphs with similar number of nodes, which shows that **N2AGim can effectively address N2AI**.
> * We propose the GSL2 framework to accelerate graph similarity inference. Experimental results show that:
> 	1. The accuracy of GSL2 can be improved by using a graph similarity model with high accuracy. **Therefore, using a model that can effectively solve N2AI to achieve higher accuracy, i.e. N2AGim, can effectively improve the accuracy of GSL2.**
> 	2. N2AGim-based GSL2 can be **up to 12.9, 11.3 and 47.7 times faster** than N2AGim.
> 	3. Compared to other models, GSL2 achieves the **state-of-the-art performance** in terms of inference speed.
>
> _We hope our responses convincingly address your concerns. Please do not hesitate to contact us if there are other clarifications or experiments we can offer_.
>
> _Thank you for your time again!_
>
> Best, Authors

---

### Official Review · Reviewer_2mHs · 2022-10-26

**Confidence:** 4
**Correctness:** 3
**Technical Novelty And Significance:** 1
**Empirical Novelty And Significance:** 1
**Recommendation:** 3

**Clarity, Quality, Novelty And Reproducibility:**

The paper is clearly written but the figures are too small and not readable.
The code is provided but I did not run it.

**Strength And Weaknesses:**

The motivation in the introduction is lost in the rest of the paper. The main claim in the introduction is that GNNs are not good for predicting GED when sizes of the input graphs are close but the empirical results do not show that the current algorithm is actually improving this situation.

Similarly, theoretical results about pooling in Section 3 are useless. Indeed, pooling is not discuss anymore in Section 4 describing the architectures used. In the experimental section (section 5), pooling is studied in the ablation study and there seem to be no clear conclusion.

The performances of GSL2 will depend on the landmarks chosen and in particular on the number M of landmarks. A discussion about the impact of M would have benn nice.

**Summary Of The Paper:**

This paper studies the task of learning graph similarity, i.e.graph edit distance (GED)  with Graph Neural Networks. The authors propose 2 algorithms: a) N2AGim is an adaptation of the architecture proposed in (Qin et al 2021) where the attention layer is now not anymore obtaine via concatenation but through the differences of the embedding resulting in a higher attention when differences are higher; b) GSL2 leverages the first algorithm by computing approximate GED thanks to N2AGim for a set of so-called landmarks and using these values as an embedding of the graph.
The paper finishes with some experiments showing that this method gives better performances.

**Summary Of The Review:**

The contribution is very weak compared to (Qin et al 2021) and should be better discussed.

---

> ### Author Response · Authors · 2022-11-12
> **Response to Reviewer 2mHs (2/2)**
>
> > Q2 : The motivation in the introduction is lost in the rest of the paper. The main claim in the introduction is that GNNs are not good for predicting GED when sizes of the input graphs are close but the empirical results do not show that the current algorithm is actually improving this situation.
>
> That's a very good question! Fig1 demonstrates that the accuracy of the GNN-based similarity models decreases as the SizeDiff, the percentage difference in the number of nodes in the two graphs, decreases, which is defined as the N2AI. To verify whether N2AGim can effectively address N2AI, we visualize the performance of different models on graph pairs with similar numbers of nodes in **Fig5** of the resubmitted paper, and provide detailed numerical comparisons in the **Appendix G.1**. For example, the comparison on AIDS700nef is shown below :
>
> | SizeDiff | H2MN  | EGSCS | GraphSim | EGSCT | N2AGim  |
> |----------------|-------|-------|----------|-------|---------|
> | \[0.000, 0.075)   | 2.880  | 2.265 | 2.485    | 2.272 | **1.661**   |
> | \[0.075, 0.150)  | 2.059 | 1.611 | 1.869    | 1.694 | **1.226**   |
> | \[0.150, 0.225)  | 1.442 | 1.260  | 1.555    | 1.428 | **0.898**   |
> | \[0.225, 0.300)   | 1.545 | 1.615 | 2.022    | 2.182 | **0.894**   |
> | \[0.300, 0.375)   | 0.916 | 0.901 | 0.917    | 1.443 | **0.573**   |
>
> Note that the graphs in AIDS700nef all have a relatively small number of nodes, and a 7.5% difference in node number can be seen as a difference of approximately one node. The results demonstrate that N2AGim achieves better performance than the other models at all levels of SizeDiff, especially about 26.7% better than the second place EGSCS (1.661 vs 2.265 on MSE) when the difference in number of nodes is less than 7.5%. This is a strong evidence that N2AGim practically improves N2AI. Please see Fig5 and Appendix G.1 in the resubmitted paper for further details.
>
>
> > Q3 : Similarly, theoretical results about pooling in Section 3 are useless. Indeed, pooling is not discuss anymore in Section 4 describing the architectures used. In the experimental section (section 5), pooling is studied in the ablation study and there seem to be no clear conclusion.
>
> Thanks for the question. **Theoretical results about pooling in Sec3 are useful, and it is one of the CORE contributions of our paper.** Indeed, the aims of the Sec3 is to provide a formal theoretical analysis of N2AI and reveal the reasons for its existence, rather than trying to find the most suitable pooling method. The conclusion is that ***all* graph pooling modules map two graphs with a similar number of nodes to similar embedding distribution.** Therefore, we focus in Sec4.1 on how to design models to address the similar embeddings that are generated by all graph pooling functions, and then propose the DiffAtt module.  In Sec5.1, we verified the huge boost from the proposed DiffAtt in all four pooling functions, which illustrates that the DiffAtt can effectively address the N2AI brought by all pooling functions. **In summary, the core of Sec3 and 4.1 is not how to select a good pooling module, but rather to analyse and address the N2AI of all pooling functions modules.**
>
> > Q4 : The performances of GSL2 will depend on the landmarks chosen and in particular on the number M of landmarks. A discussion about the impact of M would have benn nice.
>
> In fact, we have already emphasized in the first paragraph of Sec 5.1 that *we also provide additional ablation experiments on the hyperparameter selection of GSL2 , including **selecting different numbers of landmarks** and using different random seeds to select different landmarks, etc. in Appendix F(It is now Appendix G).*  Please See the Appendix G in the resubmitted paper for details.
>
> _We hope that the provided new experiments and additional explanations have convinced you of the merits of our work. Please do not hesitate to contact us if there are other clarifications or experiments we can offer._
>
> Best, Authors

---

> > ### Comment · Reviewer_2mHs · 2022-12-06
> > **no convincing experiments**
> >
> > Thank you for your answer but I do not find the new experiments convincing regarding the original motivation.

---

> > > ### Author Response · Authors · 2022-12-06
> > > **Response to Reviewer 2mHs**
> > >
> > > Thanks for your feedback!
> > >
> > > We have added new experimental data to Q2 of our previous response to you and we would now like to summarise our original motivation and the conclusions of the new experiment.
> > >
> > > **In N2AGim, our original motivation is to address the N2AI.** In order to clearly demonstrate whether N2AGim successfully addresses N2AI, we tested the MSE of different models on pairs of graphs with similar numbers of nodes and the results are reported in *Fig5*.
> > >
> > > Obviously, N2AGim is more accurate in predicting graph pairs with similar numbers of nodes, especially better about 26.7% 19.4% and 35.4% than the second better performance when the difference in number of nodes is less than 7.5% on the three datasets, respectively. **The new experimental results effectively demonstrate that N2AGim practically improves N2AI.** We are therefore confident that our new experiment is convincing regarding the original motivation.
> > >
> > > *And, we would be grateful if you could further specify any irregularities in our experiments or suggest improvements to improve the quality of the paper!*
> > >
> > > Thanks!
> > >
> > > Best, Authors

---

> > > ### Author Response · Authors · 2022-12-08
> > > **Please see our new experiments and descriptions.**
> > >
> > > Dear Sir/Madam,please see our added descriptions and experiments, we are confident that our new experiment is convincing.
> > > Thanks.

---

> ### Author Response · Authors · 2022-11-12
> **Response to Reviewer 2mHs (1/2)**
>
> We thank the reviewer for the constructive comments. We have made the corresponding modifications in the newly uploaded paper based on the suggestions.
>
> > Q1 : The contribution is very weak compared to [1] and should be better discussed.
>
> Thanks for your question. Here we provide a comparison of the contribution of our manuscript with that of [1]. Since both [1] and we have proposed two different models, below we provide a detailed comparison of the corresponding models.
> - Comparisons of the N2AGim and the Teacher model in [1] :
> 	* **[Clear theory and motivation.]** We find that, for the first time to our knowledge, the global pooling function in graph neural networks maps graphs with similar number of nodes to similar embedding distributions, reducing the separability of their embeddings, which we refer to as N2AI (See Section 3). **This is one of the core contributions of our manuscript and motivates our design of DiffAtt.**
> 	* **[Different attention mechanisms, i,e., DiffAtt(our) and EFN([1]).]** DiffAtt sums the node features to obtain each graph embedding, then enhances the difference between the two embeddings, and finally concatenates the two embeddings together (See Section 4.1). EFN, on the other hand, first uses context-based attention to generate individual embeddings for each graph, and then uses a multilayer perceptron to enhance the concatenated joint embeddings.
> 	* **[Different backbone.]** We used a residual layer and an additional FFN to enhance the node features of the GIN output (See Section 4.1), whereas [1] used a pure GIN layer.
> 	* **[Better performance.]** N2AGim achieved better results than the Teacher model in 11 of the 12 metrics (See Tab 4), especially better on the AIDS700nef about 26.9% (1.170 vs 1.601 on MSE), 1.7% (0.916 vs 0.901 on $\rho$) and 2.1% (0.672 vs 0.658 on p@10). Our ablation experiments also demonstrate that DiffAtt outperformed EFN in all 12 metrics under the same conditions (See Tab 2).
> - Comparisons of the GSL2 and the Student model in [1] :
> 	* **[Different motivation.]** [1] is to accelerate the inference of graph similarity models by specially designed distillation. In contrast, the GSL2 framework is based on the relationship between GED values to accelerate inference.
> 	* **[Plug and Play.]** GSL2 is plug and play and can be applied to any GED calculator, whereas the distillation process of [1] is not a separate module that can be plugged and played.
> 	* **[Clear theory support.]** GSL2 has a clear theory which ensures that GSL2 can be improved by increasing the accuracy of the GED calculator and by selecting better landmarks. This theory also ensures that different graph similarity models as well as regression algorithms can be selected depending on the practical problem, which greatly expands the scenarios in which the GSL2 framework can be used.
> 	* **[Better performance.]** GSL2 outperforms the student model on 10 of the 12 metrics(see Table 4), and is particularly better on MSE for each of the three datasets 4.9%(1.470 vs 1.546), 74.7%(0.074 vs 0.293) and 12.2%(0.510 vs 0.581), respectively. In addition, compared to the student model, GSL2 is 1.36, 1.22 and 1.24 times faster on three datasets(see Table 5), respectively.
>
> **In summary, we believe that our manuscript makes a significant contribution both in directions not pointed out by [1] (N2AI) and in directions pointed out by [1] (accelerated similarity inference). We therefore believe that this manuscript makes a significant contribution compared to [1].**
>
> [1]Qin C, Zhao H, Wang L, et al. Slow Learning and Fast Inference: Efficient Graph Similarity Computation via Knowledge Distillation[J]. Advances in Neural Information Processing Systems, 2021, 34: 14110-14121.

---

> ### Author Response · Authors · 2022-11-18
> **Looking forward to your post-rebuttal feedback**
>
> Thanks again for your insightful suggestions and comments. As the deadline for discussion is approaching, we are glad to provide any additional clarifications that you may need, and here we would like to highlight the contributions of this paper :
> * We found and analysed N2AI in the graph similarity model, and our conclusion is that **_all_ graph pooling modules map two graphs with a similar number of nodes to similar embedding distribution, thus reducing the separability between embeddings.**
> * We propose the DiffAtt module and the N2AGim model to address N2AI. Experimental results show that:
> 	1. DiffAtt can substantially improve performance on all four pooling modules.
> 	2. DiffAtt is superior to other attention mechanisms in terms of effectiveness.
> 	3. N2AGim achieves the **state-of-the-art performance** in graph similarity learning.
> 	4. N2AGim performs better than any other model on graphs with similar number of nodes, which shows that **N2AGim can effectively address N2AI**.
> * We propose the GSL2 framework to accelerate graph similarity inference. Experimental results show that:
> 	1. The accuracy of GSL2 can be improved by using a graph similarity model with high accuracy. **Therefore, using a model that can effectively solve N2AI to achieve higher accuracy, i.e. N2AGim, can effectively improve the accuracy of GSL2.**
> 	2. N2AGim-based GSL2 can be **up to 12.9, 11.3 and 47.7 times faster** than N2AGim.
> 	3. Compared to other models, GSL2 achieves the **state-of-the-art performance** in terms of inference speed.
>
> _We hope our responses convincingly address your concerns. Please do not hesitate to contact us if there are other clarifications or experiments we can offer_.
>
> _Thank you for your time again!_
>
> Best, Authors

---

> ### Author Response · Authors · 2022-11-23
> **Looking forward to the post-rebuttal feedbacks again**
>
> It's been more than **10** days since we submitted our response and we still haven't received your feedback.
>
> Please don’t hesitate to let us know if there are any additional clarifications or experiments that we can offer, as we would love to convince you of the merits of the paper. We appreciate your suggestions.
>
> Thanks!
>
> Best, Authors

---

> ### Author Response · Authors · 2022-12-04
> **Looking forward to your post-rebuttal feedbacks again!**
>
> It's been more than **22** days since we submitted our response and we still haven't received your feedback.
>
> Please don’t hesitate to let us know if there are any additional clarifications or experiments that we can offer, as we would love to convince you of the merits of the paper. We appreciate your suggestions.
>
> Thanks!
>
> Best, Authors

---

### Author Response · Authors · 2022-11-12
**Main updates to new submission paper**

We thank all the reviewers for the insightful comments and constructive suggestions to strengthen our work. Based on all the reviewers' comments, we have revised and updated the paper in the corresponding places. The main updates are as follows.

1. We have changed the original Appendix F to Appendix G and added subsection titles to the large number of ablation experiments within it for greater clarity of presentation.
2. In order to clearly demonstrate **whether N2AGim successfully addresses N2AI**, we have changed Fig5 to show the performance of each model when the number of nodes in the graph pair is less different, and provide a detailed numerical comparison in Appendix G.1. The results demonstrate that N2AGim is more accurate in predicting graph pairs with similar numbers of nodes, especially better about 26.7%(1.661 vs 2.265), 19.4%(0.191 vs 0.237) and 35.4%(1.103 vs 1.707) better than the second better performance when the difference in number of nodes is less than 7.5% on the three datasets, respectively. **This is a strong evidence that N2AGim practically improves N2AI.**  [2mHs,w3qo]
3. We have added ablation experiments on the backbone of N2AGim in Appendix G.2 to demonstrate the effects of different GNNs, layers of GNNs and residual connections and FFNs.  [qS53]
4. We have provided inference speeds for SimGNN-based and GraphSim-based GSL2 in Appendix G.5. It is shown that GSL2-F speeds up SimGNN by 7.7, 8.5, 73 times on three datasets, respectively, and speeds up GraphSim 6.1, 10.2 and 59 times, respectively.  [qS53]

We hope our responses below convincingly address all reviewers’ concerns. We thank all reviewers’ time and efforts again!

---

### Decision · Program_Chairs · 2023-01-20

**Decision:**

Reject

**Justification For Why Not Higher Score:**

Outline and motivation need to be carefully revised.

**Justification For Why Not Lower Score:**

N/A

**Metareview: Summary, Strengths And Weaknesses:**

Computing graph-edit-distance (GED) between two given graphs is a NP-hard problem, and GNNs have been recently introduced to do a fast computation of GED. To start with, the authors show graphs with similar number of nodes can have larger approximation errors. This motivate authors to design N2AI which consider multi-scale representation and fuse embeddings with attentions. Next, the authors proposes GSL2, which select anchor points from the training graph for further acceleration. Detailed experimental results show the proposed method is promising.

Strength
- Main idea is easy to follow.
- Proposed method is motivated with theoretical results.

Weakness
- The starting point, graphs with similar number of nodes can have larger approximation errors, is not well justified. Even after rebuttal, it is not clear how performance is more improved with similar node numbers.
- Outline of the submission needs to be careful re-organized. GSL2 is in parallel with N2AI, basically they are two irrelevant methods, the focus is not highlighted.